



# MAGARA: A Multi-Angle Geostationary Aerosol Retrieval Algorithm

James A. Limbacher[1,2] , Ralph A. Kahn[3], Mariel D. Friberg[3,4], Jaehwa Lee[3,4], Tyler Summers[3,5], and Hai Zhang[1,2]

[1]I. M. Systems Group Inc., Rockville, MD, 20850, USA
[2]National Oceanic and Atmospheric Administration, College Park, 20740, USA
[3]Earth Science Division, NASA Goddard Space Flight Center, Greenbelt, 20771, USA
[4]University of Maryland, College Park, MD, 20742, USA
[5]Science Systems and Applications, Inc., Lanham, MD, 20706, USA

*Correspondence to*: James A. Limbacher (James.Limbacher@noaa.gov)

**Abstract.** For over 40 years, the Geostationary Operational Environmental Satellite (GOES) system has provided frequent snapshots of the Western Hemisphere, with its data used for a variety of tasks ranging from weather forecasting to wildfire detection. Located on the GOES-16, GOES-17, and GOES-18 platforms, the Advanced Baseline Imager (ABI) is the first GOES-series imager that meets the precision requirements (e.g., $\geq$ 10 bits per datum) for high-quality, aerosol-related research. Here, we present a pixel-level (up to 1 km) Multi-Angle Geostationary Aerosol Retrieval Algorithm (MAGARA) that leverages the ABI instruments on the GOES-16 and GOES-17 platforms, as well as the differences in autocorrelation time-scales between surface reflectance, aerosol type, and aerosol loading. MAGARA retrieves pixel-level aerosol loading and fine-mode fraction at up to the cadence of the measurements (10 minutes), fine-and-coarse mode aerosol particle properties at a daily cadence, and surface properties under a framework that combines the unique information content in multi-angle radiances (e.g., sensitivity to aerosol type from multiple scattering-angle observations) with the robust surface characterization inherent to temporally tiled algorithms such as the MAIAC method.

We present three case studies, tiling radiances for several days over the Desert Southwest (2 cases) and the Pacific Northwest (1 case). We observed/retrieved smoke from the following major fires: the Camp Fire (November 5[th]-12[th], 2018), the Williams Flats Fire (July 29[th]-August 8[th], 2019), and the Kincade Fire (October 23[rd]-November 1[st], 2019). Because GOES-17 was not making observations during the Camp Fire, we present this as a unique case demonstrating the efficacy of the multi-angle algorithm using only a single ABI sensor. We compare MAGARA retrievals of fine-mode (FM) AOD, coarse-mode (CM) AOD, and single-scattering albedo (SSA) with coincident AErosol RObotic NETwork (AERONET) spectral deconvolution algorithm (SDA) and inversion retrievals for the same period. We also compare MAGARA results against bias-corrected NOAA GOES-16 and GOES-17 retrieved 550 nm AOD.





For the 8,443 coincidences of MAGARA and the NOAA bias-corrected product with AERONET, MAGARA (NOAA bias-corrected product) 550 nm AOD error statistics are as follows: median-absolute error (MAE) = 0.016 (0.021), root-mean-squared error (RMSE) = 0.040 (0.049), and linear correlation coefficient (r) = 0.785 (0.666). At pixel-level resolution, the disparity between MAGARA and the NOAA bias-corrected product increases substantially, with MAGARA suffering less degradation in the results, likely due to lower pixel-to-pixel noise.

We report the following over-land MAGARA 500 nm fine-mode fraction error statistics for the 384 MAGARA/AERONET coincidences with MAGARA 500 nm AOD > 0.3: MAE=0.031, RMSE=0.100, and r=0.902. Combined with the presented figures of daily averaged retrieved aerosol particle properties, this suggests that MAGARA has good sensitivity to fine-mode fraction over land, especially for smoky regions.

We also compare retrievals of MAGARA spectral single-scattering albedo with AERONET. Results suggest that a 1-parameter bias correction can substantially reduce MAGARA errors at high AOD. For the MAGARA retrieved spectral AOD > 0.5 (n=116), this bias correction reduces MAE by 65% (0.028 → 0.010), RMSE by 50% (0.030 → 0.015), and improves correlation by 0.03 (0.84 → 0.87).

MAGARA performs best in regions where surface reflectance varies over long-time scales with minimal clouds. This represents a large portion of the western half of the US, much of North-Central Africa and the Middle East, some of Central Asia, and much of Australia. For these regions, aerosol type and aerosol loading on time scales as short as 10 minutes could allow for novel research into aerosol-cloud interactions, improvements to air-quality modeling and forecasting, and tighter constraints on direct aerosol radiative forcing.

## 1 Introduction

With the Television Infrared Observation Satellite (TIROS-1) launch in 1960, weather forecasting entered the space age. Although the imager onboard TIROS-1 was only operational for a couple of months, within 10 years of the TIROS-1 launch, NASA had launched an additional 20 meteorological satellites into low Earth orbit (LEO). The first full-disk imagery from geostationary orbit was acquired from the Applications Technology Satellite (ATS-1) on December 11, 1966. Five more ATS series satellites were launched over the next 10 years, including ATS-3, which took the first true-color image from geostationary orbit. Interestingly, *Warnecke and Sunderlin* (1968) present dual views from ATS-1 and ATS-3, with images from the two stapled together only 7 hours apart, resulting in a montage of the Atlantic and Pacific Oceans. Nearly 55 years later, it is now possible to view nearly all sub-polar (within ~60° latitude of the equator) regions of the planet within a few minutes of each other.

To be designated a geostationary platform, a satellite must maintain an orbit ~35,800 km above any particular point on the equatorial belt wrapping around the Earth (0° inclination). At that altitude and inclination, the satellite is stationary relative to any point on the ground, as its orbital period matches the planet's rotational period. The GOES program officially began over 45 years ago, with the launch of GOES-1 on October 16, 1975 (https://www.nesdis.noaa.gov/news/40-years-of-



goes-the-anniversary-of-goes-1, last accessed 08/24/2022). Since the launch of GOES-1, major advances have been made to
the onboard Earth-viewing imager design, resulting in significant improvements in the number of spectral bands, spatial
resolution, temporal cadence, radiometric accuracy, geometric registration, and bit depth (i.e., the amount of information in a
given pixel of data). The latest generation of GOES satellites, designated the R-series, began with the launch of GOES-16
(i.e., GOES-R) on November 19, 2016. After a check-out period to determine that the spacecraft/instruments were operating
nominally, the spacecraft was maneuvered to 75.2° W, declared operational, and designated GOES-East on December 18,
2017 (*Schmit et al.*, 2018). GOES-17 (i.e., GOES-S) was launched in March 2018, moved to 137.2° W, and declared
operational as GOES-West in February 2019 (*Wang et al.*, 2020). Although GOES-17 suffered a partial failure of its loop
heat pipe, which resulted in severe degradation, primarily in the thermal infrared spectral bands, much of the issue has since
been mitigated. MAGARA uses none of the affected bands. Additionally, the recently launched GOES-18 satellite has now
taken over operations from GOES-17 (https://www.goes-r.gov/users/transitionToOperations18.html, last accessed
05/05/2023), which formally resolves this issue.

Each GOES-R series satellite contains an Earth-viewing Advanced Baseline Imager (ABI). ABI measures upwelling
radiance in 16 spectral bands from 0.5 km to 2.0 km spatial resolution directly above the equator at the longitude of the
spacecraft, with spatial resolution inversely proportional to the cosine of view-zenith angle. Of these 16 spectral bands, two
centered on wavelengths of 0.470 μm [blue] and 0.640 μm [red] are sensitive to light in the visible portion of the
electromagnetic spectrum. Four bands centered on wavelengths of 0.865 μm, 1.38 μm, 1.61 μm, and 2.25 μm are sensitive to
reflected solar radiation in the near-infrared portion of the spectrum. These 6 bands are known as solar reflective bands because
they measure solar light that has been reflected by the Earth's atmosphere and underlying land and ocean surfaces. Ten
additional bands measure light at longer wavelengths, in which emission by the Earth tends to dominate the observed signals.
Because these 10 longwave bands are sensitive to electromagnetic radiation at wavelengths much longer than the sub-micron
size of most aerosol particles, with dust and volcanic ash as the main exceptions, we do not use these bands in the MAGARA
aerosol retrievals, as they add minimal information content on the aerosols relative to that contained in the solar reflective
bands.

As geostationary imagers view the same region of Earth 24 hours a day, these imagers are inherently optimal for
applications that require the ability to resolve changes in a local region's environment (either surface or atmosphere) on short
timescales. The most obvious and recognizable use of these platforms is for operational weather forecasting. One of the earliest
technical reports on the feasibility of taking space-borne observations of the planet was presented in 1951 and published in
1960 (*Greenfield and Kellogg*, 1960). Additional applications include observations of volcanic eruptions via satellite (*Cochran
and Pyle*, 1978), as well as classification of volcanic eruption plume particles (*Flower and Kahn*, 2020a, 2020b; *Scollo et al.,*
2012) and determination of volcanic aerosol (*Kahn and Limbacher*, 2012) and wildfire smoke (*Junghenn Noyes et al.*, 2022)
plume properties. With regards to the MAGARA retrieval described in this manuscript, potential aerosol-related applications
include improvements made to climate modeling of aerosol direct (*Matus et al.*, 2019) and indirect effects (*Quass et al.*, 2020),





as well as improvements made to air-quality modeling (*Friberg et al.*, 2018; *deSouza et al.*, 2020). Although this manuscript describes and assesses the accuracy of MAGARA for a few small case studies, there are several other research groups developing their own aerosol retrieval algorithms for geostationary Earth-viewing imagers. This includes the MODIS dark-

target (DT) group (*Gupta et al.*, 2019; *Remer et al.*, 2020), the MAIAC group (*Lyapustin et al.*, 2018; *Li et al.*, 2019; *Wang et al.*, 2022), the GRASP team (*Li et al.*, 2020), and the NOAA aerosol team themselves (*GOES AOD Algorithm Theoretical Basis Document [ATBD]*, 2018; *Liu et al.*, 2018; *Kondragunta et al.,* 2020; *Zhang et al.*, 2020), among others. Other groups have sought to use observations of both GOES-R and GOES-S to constrain the scattering phase function of dust aerosols (e.g., *Bian et al.*, 2021).

Most existing aerosol retrieval algorithms make use of a single sensor at a given time to determine aerosol loadings and properties. *Bian et al.* (2021) used both GOES-16 and GOES-17 to constrain the phase functions of dust, but the algorithm they developed is not a fully-fledged aerosol retrieval algorithm. *Govaerts et al.* (2010) used prior generations of geostationary imagers to retrieve daily aerosol optical depths and surface BRFs. This is similar to MAGARA, but they only retrieved aerosol optical depth (AOD) once per day rather than at every imager snapshot, which is a significant limitation compared to more

contemporary algorithms (with advanced geostationary imagers). The GRASP algorithm is similar to MAGARA in terms of its ability to ingest imager data from multiple platforms at multiple times in order to constrain aerosol and surface properties; their algorithm is significantly more mature (and generalizable to different instruments/platforms; *Dubovik et al.*, 2014). One reason that MAGARA may be able to add value here is our simultaneous use of dual-view imagery combined with our exploitation of the varying autocorrelation time-scales for AOD vs aerosol type (*Sayer*, 2020). *Ceamanos et al.* (2023) explored

using 15-minute imagery from the Meteosat Second Generation (MSG) platform in order to perform 15-minute retrievals of AOD (aerosol type is assumed based on geography and retrieved aerosol optical depth). *Zhang et al.* (2013) developed a simultaneous dual-view aerosol retrieval algorithm using the prior generation of GOES imagers (GOES 11-15). Much of the logic of MAGARA follows along *Zhang et al.'s* line of thinking, including the use of MAIAC data, relative calibration between GOES-East and -West, retrieval of the surface using low aerosol loading days, and using simultaneous imagery from GOES-

East and -West to retrieve aerosol loading. MAGARA takes this a step further by directly retrieving the average surface bidirectional reflectance factor (BRF) over the course of a week (or more) in lieu of using surface reflectance ratios. Additionally, MAGARA retrieves daily particle property information about the fine-and-coarse modes as well as retrieving aerosol loading and fine-mode fraction at (up to) the cadence of the input observations.

As far as the authors are aware, there is no direct analog to MAGARA out there; but any approach that makes use of

the differing autocorrelation time-scales of surface reflectance (longest), aerosol type (long), and AOD (short), could be used to extract a significant amount of information about aerosol particle properties, especially if those algorithms exploit (via multi-sensor data fusion) the next-generation geostationary ring that is currently being assembled. The layout of this manuscript is as follows: The MAGARA algorithm methodology is presented in Section 2. Section 3 outlines three separate case studies: the Camp Fire, the Williams Flats Fire, and the Kincade Fire. Section 4 details initial comparison/validation of



NOAA/MAGARA 550 nm AOD, MAGARA fine-mode fraction, and MAGARA single-scattering albedo with AERONET sun photometers. General conclusions on MAGARA performance are presented in Section 5.

## 2 Methodology

MAGARA was initially conceived as a way to maximize aerosol information retrieval by fusing top-of-atmosphere (TOA) bi-directional reflectance factor (BRF) observations (i.e., Level 1B data) from both GOES-16 and GOES-17 onto a common grid,
tiling these observations over the course of several days to a couple weeks, and then applying knowledge of aerosols/satellite remote sensing, e.g., the varying autocorrelation time-scales of AOD, aerosol type, and surface reflectance, to develop a pixel-level (1 km at the sub-spacecraft point) aerosol retrieval algorithm to convert those TOA BRFs into information about aerosol loading and aerosol type. This section details that process as faithfully as possible. A code flow chart is presented in Figure 1, outlining the process from data download to comparison with AERONET to video presentation of aerosol properties. The
subsections within this section are presented in a manner consistent with that flow chart. An example of the output file generated by MAGARA is presented in the supplemental. All MAGARA output data used for this publication are available in the repository listed at the bottom of the manuscript.

### 2.1 MAGARA Data Preparation

Section 2.1.1 details general scene selection, data download, regridding and radiometric corrections for trace gas absorption.
Section 2.1.2 outlines how MAIAC surface BRF is ingested into the MAGARA algorithm, MAGARA's radiative transfer (RT) look-up table (LUT) is presented in Section 2.1.3, and MAGARA's initial cloud screening is described in 2.1.4.

### 2.1.1 Scene Selection, GOES L1B Data Download, Regridding, and Corrections

Prior to downloading the radiance data from NOAA, we first identify the central latitude/longitude, date range, and time range (time of day) of interest, and the size of the boxes (grid) to be created and then tiled with the GOES Level-1B (L1B) TOA
BRF data. The date range of interest is typically determined by ensuring that each pixel has at least two cloud-free days (for a given time of day) and that there is sufficient computer RAM (random-access memory). We identify the time range of interest by calculating the solar geometry for our region of interest based on time of day. The time range of interest is then set to ensure that the solar geometry does not exceed our LUT values for any pixel within our region of interest (ROI). Because satellite view angle directly determines the spatial resolution for a given GOES imager, we then determine whether it would be more
useful to use GOES-R or GOES-S as our interpolated grid. For feature recognition, higher spatial resolution would typically be preferred. But for accuracy, it makes more sense to interpolate to the coarsest grid (GOES-R for the western US), as all the fine features will not be captured by imager data that have been regridded to higher spatial resolution. For all cases presented here, data are interpolated to the standard GOES-R grid, which means that GOES-R data do not need to be spatially interpolated. GOES-R and GOES-S radiance data are then downloaded for bands 1, 2, 3, 4, 5, and 6 corresponding to the





date/times of interest. The GOES radiance and BRF data are provided courtesy of the *GOES-R Calibration Working Group* and *GOES-R Series Program* [2017].

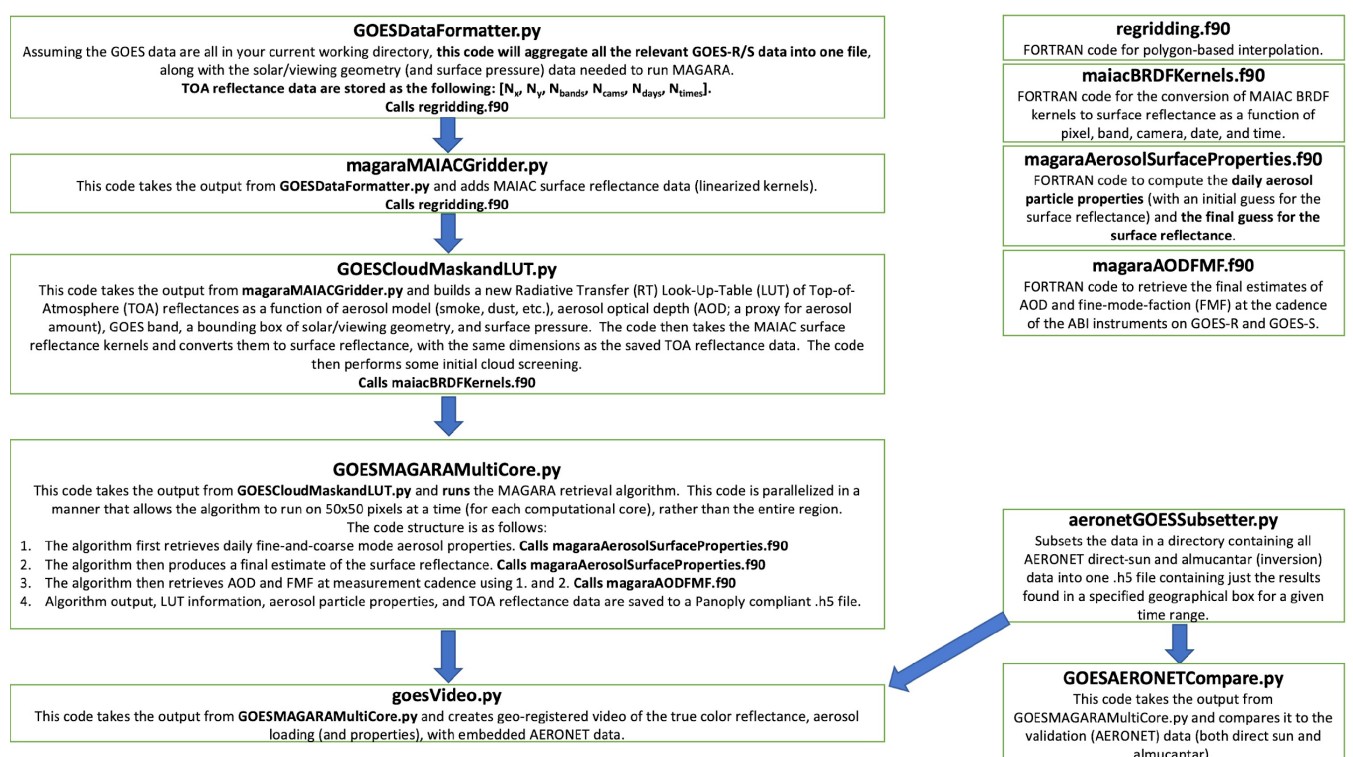

**Figure 1) Algorithmic flow chart of MAGARA, from initial data formatting to video generation and validation. Four FORTRAN**
**subroutines compiled and imported into the Python programming environment are shown in the upper right.**

Once the data are downloaded, we begin by running the Python program GOESDataFormatter.py. Because different spectral bands have different spatial resolutions (0.5-2 km), we first regrid all data to the 0.47-µm band grid (1 km at the sub-spacecraft point) either by averaging (for the 0.64-µm band) or using a bicubic interpolation for the longer wavelength, coarser

spatial resolution bands. The program, which was converted to FORTRAN 90, then regrids the GOES-S latitude/longitude grid to the GOES-R grid using the FORTRAN area-based polygon interpolation program produced by J. M. Zerzan [1989]. The final grid is established using the provided central latitude/longitude to identify the closest GOES pixel. Then the algorithm uses the user-provided bounding box size to trim the data to the region of interest. Once the regional data have been extracted, we then loop over every date/time for each band and GOES-S and GOES-R spacecraft, interpolating to the background grid,

if necessary, and converting radiances to TOA BRFs. TOA radiances are converted to TOA BRFs via the following relationship:

$$BRF = L * \frac{\pi * D^2}{E^{TOA}},$$
(1)



where L represents the observed TOA radiance (W m$^{-2}$ µm$^{-1}$ sr$^{-1}$) for a particular band and imaging platform (GOES-R or GOES-S here), D is the Earth-Sun distance at the time of observation in Astronomical Units (AU), and E$^{TOA}$ is the exo-

atmospheric solar irradiance at 1 AU (W m$^{-2}$ µm$^{-1}$). Because each spacecraft, band, date, and time corresponds to a separate radiance file (every 10-15 minutes), the algorithm must process about 3,000-10,000 radiance files for a given 7-to-14-day case study period.

After sorting and tiling these TOA BRFs into a 6-dimensional array, we correct TOA BRFs in bands 1, 2, 3, 5, and 6 for ozone and water vapor absorption. Ozone absorption primarily impacts bands 1 and 2, whereas water vapor absorption

impacts bands 3, 5, and 6 most heavily. Correcting for absorption by these gases is simplified by assuming that they are located above the significant scattering layers in the atmosphere, including those resulting from aerosols. Although this assumption is more justified for ozone, we also apply it to water vapor (with acceptable error). The following two-way transmittance correction is applied to produce adjusted BRFs that mitigate the presence of absorbing gases:

$$\mathrm{BRF_{new}} = \mathrm{BRF_{old}} * exp\left(\frac{\mathrm{OD}}{\mu_0}\right) * exp\left(\frac{\mathrm{OD}}{\mu}\right), \quad (2)$$

where the ozone (272 Dobson Units) + water vapor (1 cm) static gas absorption optical depths (ODs) are taken to be 0.0052, 0.0265, 0.0017, 0.019, and 0.0316 for the 5 bands used by MAGARA, $\mu$ represents the cosine of the viewing zenith angle, $\mu_0$ represents the cosine of the solar zenith angle, BRF$_{old}$ represents the TOA BRF prior to correction, and BRF$_{new}$ represents the TOA BRF after correction for trace gas absorption. The second factor in Equation 2 represents the correction to the solar beam as it passes through the absorbing layer, and the third factor represents the correction to the surface and atmosphere reflected

beam as it passes back up through the absorbing layer towards the satellite. The optical depths in Equation 2 are based on an atmospheric column of ozone appropriate to the region and times of interest. It is unlikely that ozone absorption varies across a region in space and time by an amount sufficient to induce significant errors. Water vapor, on the other-hand, may deviate significantly from the 1-cm column value assumed when running the RT code for the case study at hand. However, if the variation correlates only with time-of-day rather than day-to-day, the algorithm ***should*** alias the error into the retrieved surface

BRF (BRF$^{Surf}$), yielding a small bias in BRF$^{Surf}$ for the 1.6- and 2.25-µm bands for those times when column water vapor differs significantly from 1 cm. If the algorithm fails to alias the water vapor errors into BRF$^{Surf}$, they will show up in the retrievals of coarse-mode AODs, as fine-mode AODs have minimal impact on modeled BRFs at 1.6 µm and 2.25 µm. As far as we can tell, these assumptions do not negatively impact the results shown in Section 3, but dynamically varying values of ozone and water vapor would need to be ingested for any operational version of MAGARA.

Once the algorithm has corrected TOA BRFs for ozone and water vapor absorption, the algorithm regrids the data to either a 10- or 15-min time spacing, depending on GOES observation mode (or user preference), using linear interpolation. Although GOES data do not fit exactly to this 10- or 15-min cadence, these data are close to it. So, filling any temporal gaps in the data via temporal linear interpolation from the gap's bounding data is reasonable. This step does not correct for any bad data present in the radiance files, but this temporal interpolation does allow us to put the data on a regular temporal grid. These

gap-filled, corrected BRFs are then stored in a Hierarchical Data Format-5 (HDF-5) file, along with solar and viewing





geometries, latitudes, longitudes, surface pressures, days, and times for all of the data. A land/water mask is also included in the HDF-5 file for diagnostic purposes only, as the retrieval algorithm makes no distinction between land and water surfaces in the aerosol/surface retrieval process.

### 2.1.2 MODIS MAIAC Surface BRF Data Ingestion

First, MODIS-MAIAC surface BRF kernels [*Lyapustin et al.*, 2018; *Lyapustin and Wang*, 2018] are spatially interpolated to our GOES grid using the polygon-based interpolation technique described previously. This is done by running the program magaraMAIACGridder.py which spatially interpolates these surface BRF kernels for the central day of our case study period. Because these kernels contain all information necessary to produce the surface BRF at any given time of day, only one set of kernels is needed for a given band, pixel, and sensor. As MAGARA directly retrieves the average $BRF^{Surf}$ for a given band, 220 sensor, and time of day over the entire time period, the MAIAC data are needed only to correct for small changes in $BRF^{Surf}$ over that time frame due solely to small differences in solar geometry from day to day for the same time of day. Once these $BRF^{Surf}$ kernels have been spatio-temporally interpolated bilinearly to our GOES grid for every time and day on our grid, we save this information to the HDF-5 file.

Note that MAIAC only produces kernels over land. For all over water regions where such BRF kernels are missing, 225 the algorithm assumes an initial albedo and surface reflectance of 0.0 for all days, times, and spectral bands. For a given MAGARA run, the algorithm will retrieve the surface BRF of the water by aliasing differences between our aerosol model and the observations (residuals) into the retrieved surface BRF. This will give the retrieved surface BRF a blue hue over water and will not impact our retrieval of aerosol properties, as shown in our case studies.

Next, these $BRF^{Surf}$ kernels must be converted to actual $BRF^{Surf}$ values as described in *Lyapustin et al.* [2018]. As 230 shown in Figure 1, this is performed in the program GOESCloudMaskandLUT.py. These $BRF^{Surf}$ values, which are stored in the same size array as the input TOA BRFs, are then added to the MAGARA input file. As with unscreened clouds, real changes in $BRF^{Surf}$ values over a given window will cause large errors in retrieved aerosol loading and aerosol properties. For such cases, it is also likely that the algorithm will yield elevated residuals, allowing us to screen out such data.

### 2.1.3 MAGARA Aerosol Look-Up Table

Similar to the MISR research aerosol retrieval algorithm (MISR RA; *Limbacher and Kahn*, 2019; *Limbacher et al.*, 2022), and just like most other modern aerosol retrieval algorithms, MAGARA uses a prebuilt look-up table of radiative transfer outputs in lieu of running a radiative transfer code on the fly, as doing so is orders of magnitude faster. The SCIATRAN Radiative Transfer (RT) discrete ordinates solver output is corrected for atmospheric polarization effects, with ordinate density dictated by aerosol model (larger particle-size optical models use a denser grid; *Rozanov et al.*, 2014), as was done in *Limbacher et al.* 240 [2022]. The aerosol models (components) used in this paper are identical to the ones that were used in other previously published work (*Junghenn Noyes et al.*, 2020) and are presented in Table 1. All effective radii are based on an area-weighted



calculation of effective radius, as is done in Equation 5.248 of *Mishchenko et al.* [2002]. The aerosol components presented here cover the range of aerosols to which we would expect to have sensitivity with an aerosol retrieval algorithm such as MAGARA. In all sections of this manuscript, the use of the term "particle properties" is analogous to "component fraction."

This is because MAGARA actually retrieves component fraction, not particle properties, using the principle of linear mixing, meaning we assume that the TOA modeled BRFs for our aerosol components are aggregated ***linearly*** as was done with the MISR RA in *Limbacher et al.* [2022]. This linear mixing of the TOA modeled BRFs also implies that the particle properties for each individual component (e.g., SSA in Table 1) are linearly combinable to form aggregate aerosol particle properties.


**Table 1: Microphysical and optical properties of the new RA aerosol component climatology.**

| Analog (aerosol type) | $r_e$ | ANG | SSA (550 nm) | AAE |
|---|---|---|---|---|
| Very small, spherical, strongly absorbing BlS | 0.06 | 2.74 | 0.80 | 1.43 |
| Very small, spherical, strongly absorbing BrS | 0.06 | 3.17 | 0.80 | 3.23 |
| Very small, spherical, moderately absorbing BlS | 0.06 | 2.97 | 0.90 | 1.35 |
| Very small, spherical, moderately absorbing BrS | 0.06 | 3.19 | 0.90 | 3.12 |
| Small, spherical, strongly absorbing BlS | 0.12 | 1.80 | 0.80 | 1.34 |
| Small, spherical, strongly absorbing BrS | 0.12 | 2.04 | 0.80 | 3.02 |
| Small, spherical, moderately absorbing BlS | 0.12 | 2.05 | 0.90 | 1.37 |
| Small, spherical, moderately absorbing BrS | 0.12 | 2.18 | 0.90 | 3.14 |
| Medium, spherical, strongly absorbing BlS | 0.26 | 0.69 | 0.80 | 0.91 |
| Medium, spherical, strongly absorbing BrS | 0.26 | 0.76 | 0.80 | 2.36 |
| Medium, spherical, moderately absorbing BlS | 0.26 | 0.92 | 0.90 | 1.08 |
| Medium, spherical, moderately absorbing BrS | 0.26 | 0.98 | 0.90 | 2.74 |
| Very small, spherical, non-absorbing | 0.06 | 3.22 | 1.00 | N/A |
| Small, spherical, non-absorbing | 0.12 | 2.31 | 1.00 | N/A |
| Medium, spherical, non-absorbing | 0.26 | 1.22 | 1.00 | N/A |
| Large, spherical, non-absorbing | 1.28 | -0.20 | 1.00 | N/A |
| Large, non-spherical, weakly absorbing | 1.48 | -0.03 | 0.96 | 2.71 |

**BlS represents black smoke and BrS is brown smoke. Column 1 describes the aerosol analogs, column 2 represents effective radius (in μm), column 3 is the Ångström exponent (computed from 470-864 nm), column 4 is the single-scattering albedo (SSA) at 0.550-**
**μm wavelength, and column 5 is the absorption Ångström exponent (AAE, computed from 470-864 nm). Spherical aerosol component optical properties are modeled using a Mie code with an assumed lognormal particle size distribution. The nonspherical component optical models are described in Lee et al. [2017].**





The LUT dimensions presented in Table 2 are similar to the older ones used for the MISR RA. The major difference
is that the SCIATRAN RT code has to be run for a completely different set of spectral bands, as MISR and ABI do not share
similar spectral responses. Additionally, due to lower expected precision of the GOES measurements and retrievals, only 14
AOD bins are used rather than the 26 bins for the MISR RA. MAGARA uses the exact same algorithm over land and water,
thereby eliminating the need to account for whitecaps and Fresnel reflection directly in an over-water based LUT. Although
the LUT in Table 2 is relatively small, we further trim the number of bins for each dimension of the LUT prior to saving the
modified LUT directly as part of the MAGARA input file for a particular region. Over a small region, $\mu$ may only vary by 0.2-
0.5, meaning that we need fewer bins for $\mu$ in the LUT (for a given scene), rather than all 19 listed in Table 2. We also do the
same thing for relative azimuth and $\mu_0$, trimming the LUT as much as possible. For the Camp Fire case, this resulted in a 75%
reduction in LUT size, which allowed us to substantially mitigate RAM usage when running MAGARA.

**Table 2: MAGARA LUT values and dimensionality.**

| Component name (17) | 550 nm AOD (14) | λ (nm) (6) | $\mu_0$ (19) | $\mu$ (19) | $\Delta\phi$ (37) | Sfc. pressure (mb) (2) |
|---|---|---|---|---|---|---|
| sph_abs_0.06_0.80_BlS | 0.00 | 470.3 | 0.10 | 0.10 | 0 | 608 |
| sph_abs_0.06_0.80_BrS | 0.05 | 635.6 | 0.15 | 0.15 | 5 | 1050 |
| sph_abs_0.06_0.90_BS | 0.10 | 863.8 | 0.20 | 0.20 | 10 | |
| sph_abs_0.06_0.90_BrS | 0.15 | 1608.8 | 0.25 | 0.25 | 15 | |
| sph_abs_0.12_0.80_BlS | 0.25 | 2242.1 | 0.30 | 0.30 | 20 | |
| sph_abs_0.12_0.80_BrS | 0.35 | | 0.35 | 0.35 | 25 | |
| sph_abs_0.12_0.90_BlS | 0.05 | | 0.40 | 0.40 | 30 | |
| sph_abs_0.12_0.90_BrS | 0.75 | | 0.45 | 0.45 | 35 | |
| sph_abs_0.26_0.80_BlS | 1.00 | | 0.50 | 0.50 | 40 | |
| sph_abs_0.26_0.80_BrS | 1.50 | | 0.55 | 0.55 | 45 | |
| sph_abs_0.26_0.90_BlS | 2.00 | | 0.60 | 0.60 | 50 | |
| sph_abs_0.26_0.90_BrS | 2.75 | | 0.65 | 0.65 | 55 | |
| sph_nonabs_0.06 | 3.75 | | 0.70 | 0.70 | 60 | |
| sph_nonabs_0.12 | 5.00 | | 0.75 | 0.75 | 65 | |
| sph_nonabs_0.26 | | | 0.80 | 0.80 | . | |
| sph_nonabs_1.28 | | | 0.85 | 0.85 | . | |
| Dust | | | 0.90 | 0.90 | . | |
| | | | 0.95 | 0.95 | 175 | |
| | | | 1.00 | 1.00 | 180 | |

**The columns are independent of each other, with each column listing the values for the variable in the heading that are included in
the LUT. The number of values is given in parentheses at the top. The overall dimensionality of the LUT is 7.**



### 2.1.4 MAGARA Initial Cloud Screening

One of the fundamental assumptions of MAGARA is that accurate separation of the surface and atmospheric signals is possible.
This is predicated on our ability to identify days and times with clouds and thick aerosol, screen them out, and use the remaining
cloud-free days and times to retrieve $BRF^{Surf}$ for each time of day, under the assumption that the day-to-day changes in $BRF^{Surf}$
are minimal.

After trimming our aerosol RT LUT and saving it to our input HDF-5 file, we then fit a 4th order polynomial to the
TOA BRFs in the trimmed LUT for each spectral band, ABI imager (analogous to camera for MISR), pixel, and day. For the
following definition of our cost function, we assume the uncertainty in the TOA BRFs is 1% + 0.001, representing 1% relative
uncertainty plus 0.001 absolute uncertainty in the TOA BRFs, which represents a best-case scenario for either imager. For all
other cost calculations in this manuscript, we assume a more realistic value of 0.005 + 5%, as recent work suggests an
uncertainty ranging from 2-5% (*McCorkel et al.*, 2020). We then calculate the discrepancies between the fitted and observed
TOA BRFs as a cost function:

$$\text{Cost}_{d,t}^{\text{Model}} = \frac{\Sigma_\lambda \Sigma_c \left( \frac{\sqrt{w_{\lambda,c,d,t}} * \left[ \text{BRF}_{\lambda,c,d,t}^{\text{TOA}} - \text{BRF}_{\lambda,c,d,t}^{\text{Model}} \right]}{\text{Unc}_{\lambda,c,d,t}} \right)^2}{\Sigma_\lambda \Sigma_c w_{\lambda,c,d,t}}. \tag{3}$$

Here, $w_{\lambda,c,d,t}$ represents the weights given to a particular observation (1 unless the TOA BRFs are < -0.01, then 0). $\lambda$
represents the spectral band, $c$ represents the platform (GOES-16 or GOES-17), $d$ represents the day, and $t$ represents the time
of day. Unc represents the uncertainty described above, and the TOA observed ($BRF^{TOA}$) and modeled ($BRF^{Model}$) BRFs are
labeled accordingly. This cost function is then converted into a modeled weighting function via

$$\text{Weight}_{d,t}^{\text{Model}} = 1.0 - \frac{\text{Cost}_{d,t}^{\text{Model}}}{\Sigma_d \text{Cost}_{d,t}^{\text{Model}}}. \tag{4}$$

For each time of day (e.g., 10:00 UTC, 10:10 UTC, …, etc.), $Cost_{d,t}^{Model}$ identifies the days when the modeled TOA
BRFs most closely match the observations and generates a weight $\text{Weight}_{d,t}^{\text{Model}}$ bounded by 0 and 1. For times when the model
closely matches the observations, Equation 4 will yield a $\text{Weight}_{d,t}^{\text{Model}}$ near unity, whereas (4) will yield a $\text{Weight}_{d,t}^{\text{Model}}$ near
0 when the model is a poor fit to the observations. This weight is then multiplied by a temporally smoothed version of Equation
4 to capture clouds that are temporally transient. Basically, Equation 4 identifies clouds, and multiplying it by a temporally
smoothed Equation 4 captures the times around which clouds are present. This aggregate weight works well in regions where
cloud cover is minimal, or cloud cover is temporally random (California and the desert Southwest). For a given time of day, it
is entirely possible that none of the modeled TOA BRFs (for any day) fit the observations well. In this case, Equation 4 will
erroneously suggest that at least some of the observations are cloud (or aerosol) free, when this is not the case. **For this reason,
MAGARA must be allowed to run over a sufficient number of days to ensure that there are at least 2 cloud-free days
for every time of day in the dataset.** If this condition is not met, the retrieved surface BRFs for the time of day when clouds



are present every day will not be accurate. In addition, this would cause errors in the retrieval of aerosol loading and aerosol particle properties every day over the improperly weighted region.

## 2.2 MAGARA Aerosol and Surface Retrieval Description

Although the following three subsections detail the complexity of MAGARA, the overall retrieval approach is rather simple. MAGARA first performs an iterative retrieval of daily averaged fine- (components 1-15 in Table 1) and coarse- (components 16 and 17 in Table 1) mode aerosol component fractions (from which particle properties are derived), iterating between retrievals of surface BRF, time-averaged retrievals of AOD, and retrievals of daily averaged fine- and coarse-mode aerosol components. It then performs another iterative set of retrievals towards a more refined surface BRF, this time iterating only

between retrievals of surface BRF and time-averaged AOD. Finally, the algorithm performs a retrieval of FMF and ***refined*** AOD once, at the cadence of the measurements with no iteration and no time averaging. Retrieval of FMF technically means the algorithm retrieves different aerosol particle properties at the cadence of the measurements by allowing selection of different fine- to coarse-mode ratios at a 10–15-minute cadence within each day.

The order in which the final output products are generated is driven by the necessary accuracy of the input data.

Because fine- and coarse-mode components are assessed on a daily basis, whereas AOD, FMF, and surface BRF are evaluated at multiple times a day, we have many more, ranging from 30 to 50+, sets of TOA BRF measurements (and up to two ABIs) for the daily averaged fine- and coarse-mode component retrieval. As such, the required accuracy of any individual BRF is **much lower** for the daily averaged fine- and coarse-mode component retrieval than that needed to retrieve AOD or FMF, which is why the daily averaged fine-and coarse-mode component retrieval is performed first. The refined surface retrieval

follows next because the retrieved AOD and FMF are quite sensitive to errors in surface BRF. The AOD/FMF retrieval is last because we have constrained all other parameters necessary in their retrieval. The number of iterations performed in any given portion of the code was determined *empirically* by observing the required number of iterations to reach convergence (using pixel-to-pixel smoothness as an indicator of convergence) for the Camp Fire case, with the same number of iterations being applied to all three case studies.

As MAGARA is a research algorithm, we then use the same number of iterations for all three case studies below, although this will likely be refined in the future. The reason we use an iterative approach is that this is much faster than retrieving all parameters simultaneously, as explained in the supplement. For the portion of the code that performs the retrievals of daily averaged fine- and coarse-mode components along with their daily averaged fractions (Section 2.2.1), as well as for the code that performs *refined* retrievals of surface BRF (Section 2.2.2), we use different time-averages than for our FMF/AOD

retrievals. The reason behind these different time averages mostly relates to our inability to separate the atmospheric and surface scattered signals otherwise, as explained further in Sections 2.2.1 and 2.2.2.

There are only two ways to properly describe an algorithm like MAGARA: a detailed Algorithm Theoretical Basis Document (ATBD), which NASA and NOAA commonly uses for operational algorithms, or a series of flow charts. As





MAGARA is not an operational aerosol retrieval algorithm, *we present the algorithm as a series of flow charts. The next three subsections correspond to bullets 1, 2, and 3 in the algorithmic flow chart panel titled GOESMAGARAMultiCore.py within Figure 1. These bullets are further broken down as flow charts in Figures 2, 3, and 4,* with special emphasis placed on the sequential stepwise retrievals within MAGARA as opposed to retrieving all aerosol and surface parameters simultaneously.

After generating the input file as described in the previous subsections, we then run the python program GOESMAGARAMultiCore.py which splits the input TOA BRF and $BRF^{Surf}$ data into 50 x 50-pixel chunks that are (optionally) run in parallel. The Mac Pro used to generate the MAGARA output in Sections 3 and 4 ran 12 instances of these 50 x 50-pixel chunks at a time. As indicated in the algorithmic flow-chart presented as Figure 1, the algorithm first retrieves daily averaged fine- and coarse-mode aerosol component fractions in a subroutine within the file magaraAerosolSurfaceProperties.f90. The algorithm then produces a final estimate of $BRF^{Surf}$ within a separate subroutine in

that same FORTRAN file. Finally, the algorithm retrieves AOD and FMF every 10-15 minutes in a subroutine found within the FORTRAN file magaraAODFMF.f90, before saving this information in a user-friendly format, as described in the supplemental. The following three subsections expand on all aspects of the MAGARA retrieval in a manner consistent with how the algorithm runs.

### 2.2.1 MAGARA Daily Fine- and Coarse-Mode Component Fraction Retrieval

A flow-chart of this subsection, from LUT regridding to aerosol component fraction and surface retrievals, is presented as Figure 2. A common foundation of any well-built remote-sensing retrieval algorithm is the understanding that we cannot *accurately* retrieve everything. Many of the algorithm design choices in this section are based on this understanding. We emphasize that a necessary but not sufficient condition for an *accurate* particle property retrieval of daily fine- and coarse-mode component fractions is an elevated aerosol loading and multiple cloud-free views. This condition is required for the

remote-sensing measurements to contain adequate aerosol property information in the presence of a surface that reflects light, thereby typically degrading aerosol SNR. Additionally, MAGARA is only able to retrieve daily averaged fine- and coarse-mode component fractions accurately if the $BRF^{Surf}$ and AOD are *reasonably well* characterized. As explained in the previous subsection, errors in TOA modeled BRF will be larger in this step. This is acceptable because we use **many more** observations to constrain these daily averaged fine- and coarse-mode component fractions compared to either the *refined* AOD or the

*refined* surface reflectance retrievals in the subsequent subsections. To better constrain AOD and $BRF^{Surf}$, this portion of MAGARA iterates multiple times between retrievals of $BRF^{Surf}$ (for a given time of day, band, and ABI), retrievals of time-averaged AOD for a given day, and retrievals of daily averaged fine- and coarse-mode component fractions. The retrievals of


**MAGARA Retrieval of daily averaged fine- and coarse-mode aerosol component fraction**

The algorithm first takes the trimmed RT LUT of parameters and converts from solar/viewing geometry into the same dimensions as the input TOA BRF data (with the addition of AOD and aerosol component). **The following is run for every pixel in the 50 x 50-pixel region. Set iterInd=0**

LUTs for s, TT, and P are interpolated to the surface pressure of a given pixel. Initial AOD assumed to be 0.1.

**Legend**

**Aerosol/Surface Retrieval Iteration**

**Surface Retrieval**

**AOD Retrieval**

**Daily Averaged Fine- and Coarse-Mode Aerosol Component Fr. Retrieval**

Corrections for multiple reflections [1/(1-s*Albedo)] are multiplied into TT.

For all but iterInd=3, MAGARA initially assumes fine- and coarse-mode aerosol component fractions, as described in bullets a.1 and a.2 in the supplemental. The final iteration uses the results from the previous iteration as its initial aerosol model. Aggregate component fractions at low AOD are still weighted according to a.2. Interpolate aggregate daily model to this result.

**Retrieve initial surface BRF** using Equation 7

**Aerosol/Surface retrieval iteration**

Retrieval weights are updated using retrieved AOD and cost function

Retrieved surface albedo and output aerosol/surface parameters are updated.

Loop over this section 3 times. iterInd = iterInd + 1; outerInd = 0

**Aerosol/Surface retrieval iteration**

Iterate 2,3, or 5 times, for iterInd=1, 2, and 3. outerInd = outerInd + 1; innerInd = 0

Iterate 2,3, or 5 times, for iterInd=1, 2, and 3. innerInd = innerInd + 1

If outerInd >1 or innerInd>2 or iterInd ==3 then
   **Retrieve optimum Daily AOD** using only Newton's Method
else
   **Retrieve optimum AOD** on LUT coarse grid
Coarse grid retrievals are +/- 16-time bin (+/- 2.5 – 4 hour) weighted-average retrievals of AOD.

**Retrieve surface BRF** using Equation 7

If innerInd==1 then
   **Retrieve daily averaged fine- and coarse-mode aerosol component Fr.** using non-negative least squares (Eq 10). Update daily aerosol model using updated component fr.

**Figure 2)** Flow chart of *MAGARA daily fine- and coarse-mode aerosol component fraction retrieval.* Output aerosol component fractions are converted into aerosol particle properties via Table 1.






daily averaged fine- and coarse-mode component fractions are based on the principle of linear mixing. As explained in section 2.1.3, this means that TOA modeled BRFs for a given component can be weighted by a particular component fractional contribution for a given AOD, with the sum of the fractional contributions totaling unity, and added together to form one aggregate TOA aerosol mixture, as is done for the MISR RA in *Limbacher et al.* [2022]. This implies that particle properties
can be aggregated in a similar manner.

The different retrievals used by this portion of MAGARA are presented below: a retrieval of BRF$^{Surf}$, an AOD retrieval, and a daily averaged fine- and coarse-mode aerosol component fraction retrieval. The retrieval of BRF$^{Surf}$ is **very** similar to the retrievals of surface albedo (or remote-sensing BRF) presented in *Limbacher et al.* [2022]. For a given aerosol model, AOD, and BRF$^{Surf}$, we create a cost function which penalizes the difference between the observed TOA BRFs and the
modeled TOA BRFs:

$$\text{Cost}_{\lambda,c,t}^{\text{Surf}} = \frac{\Sigma_d \left[ w_{\lambda,c,d,t} * \left( \frac{\left[ \text{BRF}_{\lambda,c,d,t}^{\text{TOA}} - \left( \text{BRF}_{\lambda,c,d,t}^{\text{Path}} + \text{TT}_{\lambda,c,d,t}^{*} * \text{BRF}_{\lambda,c,t}^{\text{Surf}} \right) \right]}{\text{Unc}_{\lambda,c,d,t}} \right)^2 \right]}{\Sigma_d w_{\lambda,c,d,t}}. \tag{5}$$

Equation 5 is very similar to Equation 6 in *Limbacher et al.* [2022], with the major difference being the term with BRF$^{Surf}$, which just represents the BRF$^{Surf}$ for a given pixel's spectral band ($\lambda$), ABI platform ($c$), day ($d$), and time ($t$). The weights $w$ used here are initially the same as those presented in 2.1.4, which allows us to minimize the impact of clouds. ***One of the***
***fundamental assumptions of MAGARA is that BRF$^{Surf}$ is nearly invariant from day-to-day, for the exact same time of day***. For example, the algorithm assumes that BRF$^{Surf}$ at 17:20 UTC for the start day will be ***nearly*** the same as BRF$^{Surf}$ at 17:20 UTC on the end day. Because the BRF$^{Surf}$ model used by MAIAC allows for changes in BRF$^{Surf}$ with solar/viewing geometry, a small change in solar geometry causes a slight change in BRF$^{Surf}$ which is accounted for in $TT_{\lambda,c,d,t}^{*}$, along with accounting for multiple reflections of light off the Earth's surface:

$$\text{TT}_{\lambda,c,d,t}^{*} = \frac{\text{BRF}_{\lambda,c,d,t}^{\text{MAIAC}}}{\langle \text{BRF}_{\lambda,c,d,t}^{\text{MAIAC}} \rangle_d} * \frac{\text{TT}_{\lambda,c,d,t}}{1 - s_{\lambda,d,t} * \text{A}_\lambda}. \tag{6}$$

Here, $s_{\lambda,d,t}$ represents the average backscatter of the Earth's atmosphere, including aerosols, for all solar and viewing geometries, $A_\lambda$ represents the spectral surface albedo, and $TT_{\lambda,c,d,t}$ represents the two-way transmittance. $s_{\lambda,d,t}$ depends on time and day here because we allow the algorithm to assume/retrieve a different aerosol model (i.e., set of aerosol component fractions) for each day, and $s_{\lambda,d,t}$ varies with retrieved/prescribed AOD, which can vary with time. $\frac{BRF_{\lambda,c,d,t}^{MAIAC}}{\langle BRF_{\lambda,c,d,t}^{MAIAC} \rangle_d}$ represents the
expected fractional deviation of BRF$^{Surf}$ from the average value over the given set of days. To account for multiple reflections, we simply multiply through by 1.0 / (1.0-s$_{\lambda,d,t}$*A$_\lambda$). Similar to our prior publications, we can then ***analytically*** solve for BRF$^{Surf}$ for a given time of day by taking the derivative of Equation 5 with respect to BRF$^{Surf}$, setting the result equal to 0, and solving directly for BRF$^{Surf}$:




$$\text{BRF}^{\text{Surf}}_{\lambda,c,t} = \frac{\Sigma_d \left( \frac{w_{\lambda,c,d,t}}{\text{Unc}^2_{\lambda,c,d,t}} * \text{TT}^*_{\lambda,c,d,t} * \left[ \text{BRF}^{\text{TOA}}_{\lambda,c,d,t} - \text{BRF}^{\text{Path}}_{\lambda,c,d,t} \right] \right)}{\Sigma_d \left( \frac{w_{\lambda,c,d,t}}{\text{Unc}^2_{\lambda,c,d,t}} * \left[ \text{TT}^*_{\lambda,c,d,t} \right]^2 \right)} \quad . \tag{7}$$

The fact that Equation 7 is similar to Equation 7 from *Limbacher et al.* [2022] is no coincidence. The ABI imagers aboard GOES-16 and GOES-17 are multi-angle imagers, but we can only treat them as such if we tile the observations over time and day rather than view angle as with MISR.


Once BRF$^{\text{Surf}}$ has been retrieved for the initial assumed aerosol model and AOD (Figure 2), the algorithm iterates to a more optimal AOD. This is done by computing a cost function for aerosol loading:

$$\text{Cost}^{\text{Aero}}_{t,d} = \frac{\Sigma_c \Sigma_\lambda \left[ w_{\lambda,c,d,t} * \left( \frac{\left[ \text{BRF}^{\text{TOA}}_{\lambda,c,d,t} - \left( \text{BRF}^{\text{Path}}_{\lambda,c,d,t} + \text{TT}^*_{\lambda,c,d,t} * \text{BRF}^{\text{Surf}}_{\lambda,c,t} \right) \right]}{\text{Unc}_{\lambda,c,d,t}} \right)^2 \right]}{\Sigma_c \Sigma_\lambda w_{\lambda,c,d,t}} . \tag{8}$$

For each day and time, this cost function is calculated at every AOD bin found in Table 2, with the AOD corresponding to the

minimum cost identified as the initial guess. This value is then further refined using Newton's method. Finally, a temporally averaged AOD over ±16-time bins, or 2.5 – 4.0 hours, is calculated using the updated cloud/heavy aerosol loading weights of the AOD established via Newton's method. This weighting prevents stray clouds from impacting temporally averaged AOD, unless cloudiness is persistent throughout the day. Because this algorithm is iterative and Equation 8 is costly to compute for every AOD in our LUT, this calculation is only performed once in order to locate a general minimum in cost). Subsequently,

Newton's method is solely used to iterate towards a more optimum AOD.

The final piece of this portion of MAGARA is the retrieval of daily averaged fine- and coarse-mode component fractions. Aerosol particle properties are not directly retrieved here. Rather, we assume that the TOA BRF can be modeled as a linear combination of RT parameters together with the surface BRF, with aerosol particle properties coming from a linear combination of the properties presented in Table 1:


$$\text{BRF}^{\text{Path}}_{\lambda,c,d,t} = \sum_m \text{mixFr}_{m,d} * \text{BRF}^{\text{Path}}_{m,\lambda,c,d,t},$$
$$\text{TT}_{\lambda,c,d,t} = \sum_m \text{mixFr}_{m,d} * \text{TT}_{m,\lambda,c,d,t},$$
$$s_{\lambda,d,t} = \sum_m \text{mixFr}_{m,d} * s_{m,\lambda,d,t},$$
$$\sum_m \text{mixFr}_{m,d} = 1.0. \tag{9}$$

For all three RT parameters, we are summing over the aerosol component dimension ($m$), resulting in this dimension being

eliminated from the new parameter on the left-hand side. $\text{mixFr}_{m,d}$ represents the mixture fraction of all 17 components to the total aerosol loading for a given day. We define the fine mode as components 1-15 in Table 1 and the coarse mode as components 16 and 17 in Table 1. To solve for the optimal mixture fractions, we first set up the following system of linear equations using all TOA observations for a given day:





$$\sum_m \left[ \sqrt{\frac{w_{1,1,1}}{\text{Unc}_{1,1,1}^2}} * \left( \text{BRF}_{m,1,1,1}^{\text{Path}} + \text{TT}_{m,1,1,1}^* * \text{BRF}_{1,1,1}^{\text{Surf}} \right) * \text{mixFr}_m \right] = \sqrt{\frac{w_{1,1,1}}{\text{Unc}_{1,1,1}^2}} * \text{BRF}_{1,1,1}^{\text{TOA}}$$

$$\vdots$$

$$\sum_m \left[ \sqrt{\frac{w_{\lambda,c,t}}{\text{Unc}_{\lambda,c,t}^2}} * \left( \text{BRF}_{m,\lambda,c,t}^{\text{Path}} + \text{TT}_{m,\lambda,c,t}^* * \text{BRF}_{\lambda,c,t}^{\text{Surf}} \right) * \text{mixFr}_m \right] = \sqrt{\frac{w_{\lambda,c,t}}{\text{Unc}_{\lambda,c,t}^2}} * \text{BRF}_{\lambda,c,t}^{\text{TOA}}$$

$$\sum_m 10^9 * \text{mixFr}_{m,d} = 10^9. \tag{10}$$

For a day with 50 sets of observations (5 bands and 2 GOES platforms), this results in a weighted (using the updated weights described above) set of 501 equations and 17 unknowns. The last equation of (10) forces the fractional sum of all components to be unity, with the $10^9$ weighting ensuring that this will be the case. This system of linear equations is then solved using a non-negative least squares (NNLS) solver (*Lawson and Hanson*, 1995), as was used for the MISR RA for several years.

**2.2.2 MAGARA Refined Surface BRF Retrieval**

To this point, MAGARA has performed retrievals of daily averaged AOD, BRF$^{\text{Surf}}$, and daily averaged fine- and coarse-mode component fractions, and has yet to retrieve ***refined*** values for either AOD or BRF$^{\text{Surf}}$. As one might expect, an accurate retrieval of BRF$^{\text{Surf}}$ is critical for the accurate retrieval of AOD, and especially FMF, as described in the next subsection. Here, we describe our refined BRF$^{\text{Surf}}$ retrieval, using the retrieved daily averaged aerosol component fractions, exactly as retrieved in section 2.2.1, along with initial guesses for both AOD and BRF$^{\text{Surf}}$, as described in section 2.2.1 (and expanded upon in the supplemental). Because this algorithm uses our retrieved daily averaged fine- and coarse-mode aerosol component fractions from the previous subsection, this portion of MAGARA only iterates between retrievals of BRF$^{\text{Surf}}$ and AOD. To ***accurately*** retrieve BRF$^{\text{Surf}}$ using Equation 7, the algorithm ***first*** needs to severely penalize (weight against) times when aerosol/cloud optical depth is elevated, otherwise clouds (or heavy aerosol loading) will prevent an accurate BRF$^{\text{Surf}}$ retrieval. Therefore, we perform retrievals of AOD with no time averaging for this part of MAGARA, as this allows us to update our initial cloud weighting, improving our final retrieval of BRF$^{\text{Surf}}$.

A flowchart describing MAGARA's refined BRF$^{\text{Surf}}$ retrieval is presented as Figure 3. This section represents bullet 2 in the algorithmic flow chart corresponding to Figure 1, which explains that this refined BRF$^{\text{Surf}}$ retrieval is found within a subroutine called ***magaraAerosolSurfaceProperties.f90***. In the previous subsection we introduce three types of retrievals (i.e., AOD, daily averaged fine- and coarse-mode component fractions, and BRF$^{\text{Surf}}$) performed by MAGARA within that section of code. As stated above, because we already retrieved the daily averaged fine- and coarse-mode component fractions and use the exact component fractions from that retrieval, this section of code only uses two of those three retrievals.





**MAGARA Retrieval of refined surface BRF**

The algorithm first takes the trimmed RT LUT of parameters and converts from solar/viewing geometry into the same dimensions as the input TOA BRF data (with the addition of AOD). **Daily aerosol model is assumed from previous subsection (and flow chart).**
**The following is run for every pixel in the 50 x 50-pixel region. Set iterInd=0**

LUTs for s, TT, and P are interpolated to the surface pressure of a given pixel. **Initial AOD from previous run.**

**Legend**

**Aerosol/Surface Retrieval Iteration**

**Surface Retrieval**

**AOD Retrieval**

Loop over this section 4 times.
iterInd = iterInd + 1

Corrections for multiple reflections [1/(1-s*Albedo)] are multiplied into TT.

**Aerosol/Surface retrieval iteration**

Retrieval weights are updated using retrieved AOD and cost function

**Retrieved surface albedo and output aerosol/surface parameters are updated.**

**Aerosol/Surface retrieval iteration**

**Retrieve surface BRF** using Equation 7

Loop 5 times.
AOD time-bin average = +/- 0, 0, 0, and 32 bins, for iterInd = 1, 2, 3, and 4, respectively.

**Retrieve optimum AOD** on LUT coarse grid

**Retrieve surface BRF** using Equation 7

**If iterInd==4** perform the following AOD retrievals *with no temporal averaging*:
**Retrieve optimum AOD** on LUT coarse grid.
**Perform 5x iterations using Newton's Method** to optimize AOD.

**Retrieve surface BRF** using Equation 7

**Figure 3) Flow chart of MAGARA refined surface BRF retrieval.**

As in the previous section, this portion of the algorithm is best described as loops containing BRF$^{Surf}$ and AOD
retrievals. The initial aerosol retrieval here does not use any temporal averaging, which allows us to better identify temporally transient clouds in the scene. After a series of BRF$^{Surf}$ and AOD retrieval iterations with no temporal weights to get better





constraints on optical loading, the algorithm uses these 10-15 minute retrieved AODs to update the retrieved surface albedo and the retrieval weights, which are negatively correlated with the retrieved optical loading. The algorithm then retrieves AOD over ±32 bins, corresponding to timescales ranging from ±5 h to ±8 h, depending on the temporal cadence of the ABI

instruments during the observation window. This long-window average helps prevent extreme nonlinearity in retrieved BRF$^{\text{Surf}}$ (as a function of time of day) from aliasing into the retrievals of aerosol optical depth, which should be far less variable on days with low aerosol loading and are weighted **much** more heavily for the analytic surface BRF retrieval. After several iterations of AOD and BRF$^{\text{Surf}}$ retrievals, BRF$^{\text{Surf}}$ should have converged to a solution with minimal aerosol artifacts present. After first coarsely retrieving AOD using only the bins in our LUT with no temporal averaging, the algorithm then iterates 5

more times using Newton's method, again without any temporal averaging. Because these iterations are performed using a surface with a nearly all-day temporally averaged AOD, it is hoped, and the results over-land bear this out, that any errors present in the surface are minimal. A final retrieval of BRF$^{\text{Surf}}$ is then performed, and the ***final*** output surface parameters are updated.

### 2.2.3 MAGARA AOD and FMF Retrieval

After the previous two MAGARA steps, this portion of MAGARA now has access to ***refined daily averaged aerosol fine- and coarse-mode component fractions and a refined set of 10–15-minute BRF$^{Surf}$s***. This section now describes the portion of MAGARA that implements the ***refined*** retrieval of AOD and FMF, information that is critical to fields such as air quality. For the case studies presented in section 3, the temporal cadence of the ABI instruments ranges from 10-15 minutes. Because the retrievals of AOD and FMF presented here are performed at or near the cadence of the measurements, the temporal

resolution of these retrievals also varies between 10-15 minutes. A flowchart describing MAGARA's retrieval of AOD and FMF is presented as Figure 4. ***Note that this is different than the retrieval of daily averaged fine- and coarse-mode aerosol component fractions retrieved in section 2.2.1.*** This portion of MAGARA calls a subroutine in magaraAODFMF.f90, shown as bullet 3 in Figure 1.

    As BRF$^{\text{Surf}}$ for all times and days and daily fine- and coarse-mode aerosol component fractions have been constrained

in the prior two parts of MAGARA, this portion of MAGARA is far less complicated. For the remainder of this section, we treat each pixel, day, and time in our multi-day retrieval as a series of for (or do) loops. The algorithm first performs a retrieval of FMF, using AOD output from the code described in subsection 2.2.2 as our input AOD and daily averaged fine -and coarse-mode component fractions output from the code described in subsection 2.2.1 as our input component fractions. The algorithm then uses this retrieved FMF in order to get a final estimate of AOD.

For the retrieval of FMF described in this subsection, RT LUTs are broken down into fine (components 1-15 from Table 1) and coarse (components 16 and 17 from Table 1) modes. Using the daily averaged fine- and coarse-mode component fractions retrieved via the code described in 2.2.1, we get an initial estimate of FMF and CMF. If FMF < 0.1 or CMF < 0.1, the daily fine- or coarse-mode aerosol component fractions are too poorly constrained to use them as the initial estimate to



generate aggregate RT parameters for that particular mode. If FMF < 0.1, we modify the component fractions described in

section 2.2.1, adding equal proportions of components 7 and 8 until the FMF reaches 0.10. Coarse-mode contributions are then

reduced by the same fraction until it reaches 0.90. If CMF < 0.1, we modify the component fractions described in section 2.2.1,

adding equal proportions of components 16 and 17 until the CMF reaches 0.10. The fine-mode contribution is then reduced

by the same fraction until it reaches 0.90. We then retrieve fine-mode fraction and coarse-mode fraction under the assumption

that the fine-and coarse-mode TOA BRFs are combinable as a system of linear equations:

---

**MAGARA Retrieval of AOD and FMF**

The algorithm first takes the trimmed RT LUT of parameters and converts from solar/viewing geometry into the same dimensions as the input TOA BRF data (with the addition of AOD). **Daily aerosol model and surface parameters are assumed from previous subsections (and flow charts). The following is run for every pixel in the 50 x 50-pixel region.**

LUTs for s, TT, and P are interpolated to the surface pressure of a given pixel. **Initial AOD from previous run.**

Corrections for multiple reflections [1/(1-s*Albedo)] are multiplied into TT.

Using aerosol component fractions output from section 2.2.1, **modify** fine-and-coarse mode aerosol component fractions if FMF < 0.1 or CMF <0.1. Add equal proportions of components 7 and 8 until FMF = 0.1. Add equal proportions of components 16 and 17 until CMF = 0.1.

Renormalize component fractions and generate fine-and-coarse-mode TOA model reflectances as shown in Equation 11.

**Retrieve Fine-mode-Fraction** using Equation 11, calculating modeled reflectances for the fine-and-coarse mode fractions **for each day/time**. Update output component Fr. for each day/time.

**Retrieve optimum AOD** on LUT coarse grid **for each day/time.** Use Newton's Method to further refine AOD. **No temporal averaging**

**Calculate output cost function (Eq 8) and model output for each day/time.**

Iterate over every day and time in the dataset.


**Figure 4) MAGARA AOD and FMF retrieval flow chart.**





$$\mathrm{BRF}^{\mathrm{Fine}}_{\lambda,c,t,d} = \sum_{m=1}^{15} \left[ \sqrt{\frac{w_{\lambda,c,t,d}}{\mathrm{Unc}^2_{\lambda,c,t,d}}} * \left(\mathrm{BRF}^{\mathrm{Path}}_{m,\lambda,c,t,d} + \mathrm{TT}^*_{m,\lambda,c,t,d} * \mathrm{BRF}^{\mathrm{Surf}}_{\lambda,c,t}\right) * \mathrm{mixFr}_{m,d} \right],$$

$$\mathrm{BRF}^{\mathrm{Coarse}}_{\lambda,c,t,d} = \sum_{m=16}^{17} \left[ \sqrt{\frac{w_{\lambda,c,t,d}}{\mathrm{Unc}^2_{\lambda,c,t,d}}} * \left(\mathrm{BRF}^{\mathrm{Path}}_{m,\lambda,c,t,d} + \mathrm{TT}^*_{m,\lambda,c,t,d} * \mathrm{BRF}^{\mathrm{Surf}}_{\lambda,c,t}\right) * \mathrm{mixFr}_{m,d} \right],$$

$$\mathrm{BRF}^{\mathrm{Fine}}_{1,1,t,d} * \mathrm{FMF}_{t,d} + \mathrm{BRF}^{\mathrm{Coarse}}_{1,1,t,d} * \mathrm{CMF}_{t,d} = \sqrt{\frac{w_{1,1,t,d}}{\mathrm{Unc}^2_{1,1,t,d}}} * \mathrm{BRF}^{\mathrm{TOA}}_{1,1,t,d},$$

$$\vdots$$

$$\mathrm{BRF}^{\mathrm{Fine}}_{\lambda,c,t,d} * \mathrm{FMF}_{t,d} + \mathrm{BRF}^{\mathrm{Coarse}}_{\lambda,c,t,d} * \mathrm{CMF}_{t,d} = \sqrt{\frac{w_{\lambda,c,t,d}}{\mathrm{Unc}^2_{\lambda,c,t,d}}} * \mathrm{BRF}^{\mathrm{TOA}}_{\lambda,c,t,d},$$

$$10^6 * \mathrm{FMF}_{t,d} + 10^6 * \mathrm{CMF}_{t,d} = 10^6. \tag{11}$$

Here the weights *w* are independent of the cloud screening done initially and updated as described above. Instead, *w* = 1 if the TOA BRFs are greater than -0.01, and 0.0000001 otherwise, which eliminates unphysically dark observations sometimes found at longer wavelengths over water. Although it appears as though Equations 11 are using 6 or 11 equations to solve for 2 unknowns, CMF is just 1.0 - FMF. As such, Equations 11 are really only solving for FMF, as one could argue Equations 10 are solving for only 16 unknowns, not 17. Once we retrieve FMF and update the output aerosol models, now as a function of day and time rather than just day, we then retrieve a crude estimate of AOD using the cost function we present as Equation 8, calculated for all AOD bins and using the weights described a few sentences above. First and second derivatives of cost with respect to AOD are then computed, and Newton's method is used to iterate to the final reported AOD at the cadence of the ABI measurements. A 10–15-minute new cost function value is then produced, which represents the cost presented in the final product.

### 2.3 The AERosol RObotic NETwork (AERONET)

In section 3 we use AERONET measurements and retrievals from dozens of sites across the western half of the United States to validate AOD and aerosol particle properties for our three case studies. AERONET sun photometers directly measure spectral AOD (*Holben et al.*, 1998) at an uncertainty of ~0.01 (*Eck et al.*, 1999; *Sinyuk et al.*, 2012). As in *Limbacher et al.* [2022], we first interpolate AERONET V3 L1.5 AOD to the MAGARA band centers, using a second-order polynomial in log-space (*Giles et al.*, 2019; *Sinyuk et al.*, 2020). We attempt to limit spatio-temporal variability from negatively impacting MAGARA to AERONET comparisons by masking out all AERONET data falling outside a ±10/15-minute window centered on the GOES acquisition time. AERONET AODs for each of the spectral bands are then averaged over this window prior to comparison with MAGARA, as well as for comparison to the NOAA GOES-16 operational AOD product.

O'Neill *et al.* [2003] developed a spectral deconvolution algorithm (SDA) to convert AOD at multiple wavelengths into a parameter related to fine-mode fraction and total aerosol optical depth at a 500-nm wavelength. The idea behind the algorithm is that fine-mode AOD should be highly sensitive to wavelength, because fine-mode aerosols are generally smaller than the wavelengths of light used by sun-photometers. This causes fine-mode AOD to drop, often dramatically, as wavelength



increases, generally reaching undetectable values out near 2.25 μm, the longest wavelength used in MAGARA. Coarse-mode AOD often behaves differently with wavelength, as it may show little change in AOD with increasing wavelength. One of the most useful features of SDA retrievals is that they are performed at the 15-minute or so cadence of the AERONET direct-sun measurements, rather than at the cadence of the almucantar inversions, which are hourly and performed only under favorable scattering geometries. In sections 4.1 and 4.2, we provide comparisons of MAGARA retrieved 500 nm AOD, and 550 nm FMF with the same parameters (FMF at 500 nm) as retrieved by AERONET using their spectral deconvolution algorithm. As previously mentioned, MAGARA fine-mode fraction is defined as fractional 550- nm extinction due to aerosol components 1-15 in Table 1.

We also compare SSA to AERONET almucantar retrievals of SSA (*Dubovik and King*, 2000), with interpolation to 470 nm, 550 nm, 636 nm, and 864 nm done in an identical manner as above. Instead of averaging AERONET inversion data, we take the temporally closest MAGARA time, if that closest time is within ± 10-15 minutes of any GOES-16/17 observation. Although AERONET almucantar inversions are retrievals themselves, they provide an opportunity to assess, or at least compare, particle properties retrieved from imagers such as ABI. Although we present all AERONET and MAGARA SSA coincidences where MAGARA AOD is greater than 0.3, it is important to note that AERONET SSA uncertainty itself increases with decreasing AOD.

## 2.4 MAGARA Cloud Screening

MAGARA cloud/quality screening consists of applying 7 separate thresholds for a given pixel, day, and time, with any indicating the presence of cloud labelled as bad data.

1. AOD<0.05 and cost function > 1,
2. A change in coarse-mode AOD of > 0.05 from one time step to the next over land or >0.10 over water,
3. Retrieval cost function > 10,
4. Daily minimum cost function > 0.5,
5. Temporal minimum cost function > 0.5,
6. Daily minimum cost function (for a given pixel) > 3 times the 68th percentile daily minimum cost function calculated over the entire retrieval region and cost > 0.5,
7. The minimum (among GOES-R or GOES-S ABI) relative spectral standard deviation (standard deviation/mean) of TOA BRF for ABI bands 1-3, 5, and 6 < 0.2 (Over water only).

Failing any of these 7 tests automatically sets the pixel QA value to 1. Afterwards, a pixel may be reclassified as good if the minimum 470 nm/2.25 micron BRF ratio exceeds 3 (strongly indicating smoke/pollution). Afterwards, a 3x3 spatial maximum filter is applied to these QA data, and a rolling maximum temporal filter (+/- 3 time steps) is applied. The tests and threshold values described above were devised empirically, by looking at obvious meteorological clouds and surface artifacts in the



output MAGARA data. Although these tests appear to work well for the three case study fires (and the hundreds of millions of retrievals therein), further work is required to assess their performance in a range of additional scenes.

The 1st test screens for cloud shadows by identifying regions where the retrieved AOD is low and the model fits are poor. Because clouds are comprised of coarse-mode droplets, any rapid increase in coarse-mode AOD could be interpreted as a cloud (2nd test). The 3rd test is a simple goodness-of-fit test. The 4th, 5th, and 6th tests are acknowledgements that if the fundamental assumptions underlying MAGARA are violated, the results will be poor. Namely, if cloud cover is consistent from day-to-day for the same time of day, MAGARA's initial cloud screening will fail to identify all days as cloud. This **could**

negatively impact the retrievals of aerosol for both that time of day, and for the entire day itself, depending on how many cloud-free retrievals were produced for that day. This test also tends to mask out regions where the surface BRF is not temporally stable. The last over-water-only test is based on the principle that liquid clouds are nearly spectrally invariant over MAGARA's spectral range, hence they look white, whereas aerosols in the size range typically observed in the atmosphere show greater spectral variability. The spatial and temporal filters applied afterwards just ensure that we don't miss the edges

of clouds.

### 2.5 NOAA GOES-16/17 Bias Corrected Aerosol Product

The operational GOES ABI AOD retrieval algorithm retrieves AOD at 550 nm from GOES-East and GOES-West (ABI AOD ATBD, 2018) L1b data. The algorithm also yields a data quality flag (DQF) with the following possible values: 0 (high quality), 1 (medium quality), 2 (low quality), and 3 (no retrieval). The retrieval algorithm over land is based on the assumption of linear

relationships between the surface reflectance at 0.47 μm, 0.64 μm, and 2.2 μm, derived from AERONET and ABI high quality data. The uncertainty in these surface reflectance ratios can cause diurnal biases in retrieved AOD, especially for low-medium quality data. Therefore, a bias correction algorithm was developed to reduce the bias (Zhang et al., 2020). The bias correction algorithm derives AOD bias by analyzing the 30-day period surrounding a given time and looking for the lowest AOD. A diurnal curve is fitted to obtain the lower bound of the 30-day AOD. The difference between the lower bound and the

background AOD is assumed to be the bias. The bias-corrected AOD is obtained by subtracting the bias from the original AOD.  Validation shows that the bias correction can improve the performance for the medium-high quality AOD retrieval: correlation increases from 0.87 to 0.91, mean bias decreases from 0.04 to 0.00, and RMSE decreases from 0.09 to 0.05. Additionally, the medium quality bias-corrected retrievals improve significantly enough to warrant use in quantitative applications, resulting in a nearly 100% coverage increase. As a result, we use all bias-corrected results with a DQF <= 1.

Although NOAA produces a 2-km (1 km for MAGARA) L2 AOD product for both GOES-R and GOES-S, the GOES-S version was not available for the duration of the Camp Fire. As a result, Camp Fire comparisons between MAGARA and NOAA GOES AOD were performed using the bias corrected GOES-16 AOD product. For the two other case studies, we make use of the bias-corrected GOES-17 AOD product, which provides substantially more high-quality retrievals (due to the lower view zenith angle for GOES-17). Because MAGARA was run at 15-minute cadence for the Kincade Fire case (using 10-





590 minute input BRFs), the NOAA bias corrected aerosol product was first regridded to the 15-minute cadence of MAGARA via linear interpolation (temporally) for AOD, and making use of a maximum filter for the DQF.

## 3 MAGARA Case Studies

This section outlines three separate case studies that demonstrate MAGARA capabilities and provide an initial assessment of MAGARA performance. AERONET validation for all three case studies are aggregated and presented in Section 4.

595 ### 3.1 Camp Fire, Desert Southwest Region (11/05-11/12, 2018)

The 2018 Camp Fire, California's deadliest wildfire in history *(Erin Baldassari,* 2018), killed 85 people and burned > 150,000 acres of north-central California from November 8-25, 2018 (*Maranghides et al.,* 2021). The fire was entirely contained within the county of Butte, California (https://www.fire.ca.gov/incidents/2018/11/8/camp-fire/, last accessed 09/03/2022), and caused the destruction of the town of Paradise, along with over 16 billion dollars of damage (*Alejandra Reyes-Velarde*, 2019). A

600 snapshot of the Camp Fire, with MAGARA retrievals of aerosol properties and surface BRF is presented in Figure 5. Videos can be generated from the MAGARA output for the 3 case-studies, although this is not presented here.

 Although the Camp Fire began on November 8th, MAGARA was run from November 5-12th, as the low optical loading days at the start of the time-period were necessary for the algorithm to derive an adequate constraint on BRF$^{Surf}$. This MAGARA run is also unique in that it was run **solely** with GOES-R, as GOES-S data were not yet available. During this

605 period, GOES-R ABI full-disk cadence was 15 minutes, and MAGARA results (and output data) are presented here at that cadence as well. The reason that we specifically centered on this region, rather than slightly further north, is due to the large density of AERONET sites found in the selected area (and the presence of some dust plumes).

 Daily-averaged retrievals of AOD, FMF, effective-radius, SSA, as well as a time-averaged RGB image are provided for context as Figure 6. Looking at the entire region, MAGARA retrieves almost all smoke plumes observed as fine-mode-

610 dominated aerosol. Not only does this match expectation, but we can confirm this with AERONET retrievals of fine-mode fraction over the same region. According to Figure 6, retrieved aerosol single-scattering albedo appears suppressed on November 9th and 10th for California's Central Valley. Although there are a few AERONET sites in this region that retrieve SSA, we can confirm that significant absorption is present by comparing the TOA BRFs for a ~10,000$^2$ km region in the northern central valley for the same time of day (10:30 AM) over the case-study observation period.

615 We present these spatially-averaged BRFs as Figure 7, with the visible bands shown in the upper panel and the NIR bands shown in the lower panel. All GOES-R bands are relatively stable from November 5th-8th, but the visible bands show a large BRF increase on the 9th as the Camp Fire plume passes over the region. All NIR bands show a BRF decrease on the 9th and 10th, with larger decreases at shorter wavelengths. The reason is twofold: most of these absorbing aerosols tend to be **very** absorbing in the NIR, and most natural land surfaces are extremely bright in the NIR (> 800-nm wavelength), especially at

620 870 nm and 1.6 μm. When an absorbing aerosol resides over a bright surface, it will diminish the brightness of the scene. For



**Figure 5) Context daily-averaged MAGARA retrievals over the Camp Fire region, Southwest United States, November 8th-12th, 2018. All panels, including the RGB images, are time-averaged for the entire day. Weights are also applied (0.0001 if not QA, 1 if the retrieval passes QA) prior to the time averaging. Each row corresponds to a separate day (8th for the top row and 12th for the bottom row). The first column represents the weighted, temporally averaged RGB image for the entire day. The second column represents the weighted, temporally averaged 550 nm AOD. The third column represents the weighted, temporally averaged 550 nm fine-mode fraction. The fourth column represents the weighted, temporally averaged effective radius (in microns). The fifth column represents the weighted, temporally averaged 550 nm single-scattering albedo. AOD and particle property results are shown only if the temporal weight sum is > 1 for the day (at least 1 valid retrieval). Otherwise, they are masked as grey. Particle properties are only shown if AOD > 0.30, which represents the minimum we show for comparison to AERONET as well.**





a given aerosol absorption, a brighter surface will yield a larger decrease in TOA BRF. For this region, the largest absorbing aerosol loading is retrieved on 9 November and 10 November. Even with this enhanced absorption, scattering dominates in the blue and red bands. However, at 870-nm and 1.6-μm wavelengths the TOA BRF drops significantly due to the absorption of these aerosols. Because these aerosols are fine mode dominated, hence small, the signal drop is largest at 870 nm, due to a

much higher spectral AOD (even though SSA is likely higher), with the decrease significantly mitigated at 2.25 μm due to a substantially lower spectral AOD and a darker surface. Over-water, $BRF^{Surf}$ at 870 nm, 1.6 μm, and 2.25 μm is nearly 0, except in sun glint, as is the signal due to minimal Rayleigh scattering, meaning we will not see any loss in signal, only increases in TOA BRF due to scattering.



**Figure 6)** Line plot of the GOES-16 10:30 AM PST (17:30 UTC) over-land TOA BRFs in the north-central Camp Fire region as a function of day. For this region and time of day, the smoke plume is observed beginning on 9 November 9, with aerosol absorption decreasing every day afterwards.




### 3.2 Williams Flats Fire, Pacific Northwest Region (07/29-08/08, 2019)

The Williams Flats Fire began on 2 August 2019 and burned nearly 45,000 acres of forest, grass, and brush in the northeastern region of Washington State. During this time, NASA and NOAA were conducting a joint field campaign in the western United States designated the "Fire Influence on Regional to Global Environments and Air Quality" (FIREX-AQ). As a result of this field campaign, NASA provided AERONET sun photometers across the region to measure total column optical loading. There were many other instruments involved, including NASA's Cloud Physics Lidar (CPL), which flew on

NASA's ER-2. The MAGARA dataset presented in this section was the same dataset used to help constrain the lidar ratio (i.e., the ratio of extinction to backscatter) in Midzak et al. [2022]. Although the Pacific Northwest is not an ideal region for satellite aerosol remote sensing due to persistent cloudiness, especially for an algorithm that requires both stable surface reflectances and non-persistent cloud-cover, MAGARA was run for the time frame of July 29–August 8, 2019. For the case study presented here, the output 10-minute cadence of MAGARA matched the input 10-minute cadence of the GOES-ABI

full-disk imagery.

We present daily-averaged aerosol particle property results for the Williams Flats Fire region from August 3rd-August 8th, 2019, as Figure 7. Unlike the other case studies presented here, MAGARA struggled with the persistent cloud cover found in parts of the William Flats region. Additionally, the surface BRF was not stable for a region centered on 47º North, 117º West. Looking at this region in Figure 7, one can observe the "no retrieval" hole increasing in size with time due

to this change in surface BRF (and the algorithm's inability to adapt to it). Even though we were able to screen out these retrievals, this reveals another limitation of MAGARA. Although our good QA retrievals agree well with AERONET here, AERONET was unable to capture any major smoke plume, as AERONET-observed AODs never exceeded 0.5 even though MAGARA daily-averaged AOD exceeded 2 on the August 8th. Compared to the previous Kincade and Camp Fires, retrieved single-scattering albedos were much higher for the Williams Flats Fire, with the exception of very near source retrievals.

Figure 8 shows an interesting comparison of retrieved single-scattering albedo vs distance from a 2.25 micron hotspot for all retrievals found within the Williams Flats Fire region. Although the smoke plumes are relatively absorbing near the source, the smoke rapidly becomes nonabsorbing within about 20 km from the source. Tiling observations over a day, an algorithm such as MAGARA can extract enough information from ABI to observe changes in aerosol particle properties over short distances. The SSA results presented here are similar to retrievals from the information-content-rich

MISR research algorithm (*Junghenn Noyes et al*., 2020), which was used to study the same plume.



**Figure 7) Same as Figure 5, but for the Williams Flats Fire region, Pacific Northwest, August 3rd-8th, 2019.**

## Single-scattering albedo vs distance from 2.3 micron hotspot for the Williams Flats Fire| 2.3 micron BRF > 0.99; Cost<20.00


**Figure 8) Comparison of average 550 nm single-scattering albedo (y-axis) vs distance from a 2.25 micron hotspot (x-axis) with a BRF > 0.99 for the Williams Flats Fire region, Pacific Northwest, August 3rd-8th , 2019. Averaging was done in AAOD and AOD space, prior to converting to single-scattering albedo, as this will minimize the impact of retrievals outside the smoke plume. In order to minimize the impact of clouds, only retrievals with cost < 20 were used in this analysis.**


### 3.3 Kincade Fire, Desert Southwest Region (10/23-11/01, 2019)

The Kincade fire began on 23 October 2019, burning nearly 78,000 acres of land in Sonoma County, California, through 6 November (Cal Fire, 2020). Interestingly, GOES-17 detected the heat signature from this fire within a minute of detection from ground-based cameras (Lindley et al., 2020). Figure 9 presents daily-averaged retrievals of AOD, FMF, effective radius,

and SSA for the Kincade Fire region from October 24-29, 2019, even though MAGARA was run from October 23-November





1. Like the Camp Fire case study, MAGARA was run for several relatively clean days to retrieve BRF$^{Surf}$. Unlike the Camp Fire case, where the temporal cadence of the input observations was 15-minutes, the temporal cadence of the input observations here is 10 minutes. The output cadence of the MAGARA retrieval here is 15 minutes, meaning the algorithm automatically interpolated the data to a coarser temporal grid. Although this was initially unintentional, this ended up being a useful way to

ensure that the temporal interpolation code worked as intended. So we present the results (and the output data) for this section using that 15-minute temporal grid.

As in the previous cases, MAGARA identifies almost all smoke plumes as either fine-mode dominated, or a mixture of fine-and-coarse aerosol. Retrieved effective radius falls within the range of expectation for biomass burning (~0.1-0.2 microns), with 550 nm SSA varying from 0.8-0.9 on the west coast, to 0.9-1.0 for the fire in Arizona. On October 27[th]

MAGARA retrieves significant coarse mode over the southern Central Valley region of California. Visual inspection of the TOA BRFs (not shown), and a comparison with AERONET (shown in the comparison section), indicate that this was a dust plume activated by a frontal system moving from the north to the south. The plume was present in diminished quantity the following day, but only a few pixels of retrievals passed our QA threshold. Another dust plume was activated over the southern Central Valley on October 30[th], but a radiometric anomaly on the GOES-16 ABI significantly impacted our retrievals for this

day (the anomaly caused the georegistration to be off by several dozen kilometers for one time-step). Because MAGARA tiles observations for all bands and ABIs over time and day, one bad set of BRFs can negatively affect the retrievals for the entire day or possibly even the entire period. This represents one of the significant limitations of MAGARA, as it makes large-scale data processing very difficult. Regardless, properly identifying both fine-and-coarse-mode aerosol in the same scene over land represents an important step for an algorithm that uses only scalar, single-view-angle BRFs (albeit from different imagers and

for different times).



mg_ref id="2" />



**Figure 9) Same as Figure 5, but for the Kincade Fire region, Southwest United States, October 24th-29th, 2019.**





## 4 MAGARA AERONET comparison and validation

Here we present comparisons of MAGARA AOD and the NOAA G16/17 bias-corrected ABI AOD retrievals with coincident AERONET data. We also compare retrievals of MAGARA and AERONET FMF and SSA. An overview of the optimal spatial averaging window to use for comparison with AERONET is presented in the supplemental.

### 4.1 MAGARA and NOAA G16/17 bias corrected AOD, and comparison with AERONET

Figure 10 shows 2-D histograms of MAGARA vs. AERONET AOD and 2-D histograms of the NOAA bias corrected
product vs AERONET for the 9x9 spatial averaging window identified in the supplement. For the 10,148 MAGARA/AERONET coincidences, MAGARA presents a RMSE of 0.062, MAE of 0.019, correlation coefficient (r) of 0.903, and a fairly large bias of 0.017. For the 10,431 NOAA bias corrected/AERONET coincidences, error statistics are as follows: RMSE = 0.057, MAE = 0.023, r=0.644, and a small bias of 0.005. For the 8,443 MAGARA/NOAA coincidences where both aerosol retrieval algorithms provide at least 1 quality assessed retrieval over the 9x9 region centered on
AERONET, statistics are as follows: RMSE=0.040, MAE=0.016, r=0.785, and a bias of 0.011 for MAGARA, and RMSE=0.049, MAE=0.021, r=0.666, and a bias of -0.002.





**Figure 10)** 2-D histograms of MAGARA and NOAA bias corrected 550 nm AOD retrievals vs AERONET 550 nm AOD. The upper left panel represents the MAGARA vs AERONET 550 nm AOD 2-D histogram with a 9 x 9 pixel spatial average. The upper right panel represents the NOAA GOES-16/17 bias corrected vs. AERONET 550 nm AOD 2-D histogram with a 9 x 9 pixel spatial average. The bottom rows represent the same as the top row, but with the requirement of at least 1 valid MAGARA and NOAA bias corrected pixel for each AERONET coincidence (apples-to-apples). A standard +/- (0.03 + 0.15 * AOD) uncertainty envelope is provided for reference, with the percent meeting that threshold indicated. Statistics for each panel are presented in the title.

## 4.2 MAGARA fine-mode fraction comparison with AERONET

Similar to our MAGARA/NOAA comparison, but now only for MAGARA, we first identify all MAGARA/AERONET SDA coincidences within +/- 15 minutes (10 for the Williams Flats case). We then save 9x9 pixels of MAGARA spectral AOD and fine-mode fraction centered on the AERONET site, as well as AERONET 500 nm fine-mode AOD and



AERONET 500 nm coarse-mode AOD. After identifying all good data (over land and water), we spatially average all

MAGARA data for each AERONET coincidence prior to spectrally interpolating the MAGARA data in log-log space to 500 nm. This allows us to compare 500 nm AOD and 550 nm fine-mode fraction with AERONET 500 nm AOD and 500 nm fine-mode fraction. The difference in wavelength for the fine-mode fraction comparison is likely much less significant than the difference in fine-mode definition for MAGARA vs. AERONET. Therefore, MAGARA 500 nm fine-mode AOD is presented as 550 nm FMF multiplied by 500 nm spectral AOD. MAGARA 500 nm coarse-mode AOD is presented as (1.0 –

MAGARA 550 nm FMF) multiplied by 500 nm spectral AOD.

        Figure 11 presents an over-land comparison of MAGARA /AERONET total 500 nm AOD, MAGARA 550 nm fine-mode fraction vs. AERONET 500 nm fine-mode fraction, MAGARA 500 nm fine-mode AOD vs. AERONET 500 nm fine-mode AOD, and MAGARA 500 nm coarse-mode AOD vs. AERONET 500 nm coarse-mode AOD. The upper-left panel of Figure 11 is very similar to the upper-left panel of Figure 10, with the only difference being that Figure 11 is plotted

as a function of 500 nm AOD instead of 550 nm AOD. The slight discrepancy in the number of coincidences is due to Figure 10's requirement of at least $51^2$ MAGARA retrievals centered on the AERONET site, vs Figure 11's requirement of $9^2$; meaning we are including AERONET sites in Figure 11 that were not present for the spatial averaging window analysis presented in the supplement. Overall statistics for Figure 11 are presented in the title of each panel, but statistics for the 384 coincidences of fine-mode fraction with MAGARA 500 nm AOD > 0.3 are as follows: MAE = 0.031, RMSE = 0.10, and

correlation coefficient = 0.902.

        An over-water comparison of MAGARA and AERONET 500 nm AOD (and FMF) is provided as Figure 12. Note that there are far fewer coincidences of MAGARA vs AERONET, and most of the coincidences should be farther from the AERONET site (over ocean). This means that it is more likely that MAGARA reports a smoke plume when AERONET observes clear sky, as seems to be indicated by the lower-left panel of MAGARA/AERONET 2-D histograms of

fine-mode AOD. Because the AERONET sites closest to water don't appear to observe as much coarse mode aerosol as sites further inland, the FMF statistics over water can't be used to identify smoke/dust discrimination. The statistics for the 117 over-water coincidences with 500 nm MAGARA AOD > 0.3 are as follows: MAE = 0.045, RMSE = 0.088, and r = 0.783. Interestingly, the clear high bias in MAGARA retrieved fine-mode AOD appears somewhat balanced by a low bias in MAGARA retrieved coarse-mode AOD. This discrepancy could be either algorithmic (we don't use a roughened ocean

surface model at all for MAGARA) or just due to smoke/dust aerosol variability discrepancy for these case studies. The Kincade and Camp Fire plots of daily averaged particle properties indicate that we should have good sensitivity to FMF over water. Still, we need more data in order to quantify this. Regardless, our total AOD error statistics (MAE = 0.013, RMSE = 0.087, r=0.889) are similar to our over-land retrievals.



**Figure 11) Over land comparison of MAGARA vs. AERONET AOD, fine-mode fraction, fine-mode AOD, and coarse-mode AOD. The upper-left panel represents a 2-D histograms of MAGARA vs AERONET 500 nm AOD. The upper-right panel is a scatterplot of MAGARA 550 nm fine-mode fraction vs AERONET 500 nm fine-mode fraction, for MAGARA retrieved 500 nm AOD > 0.3. The lower-left panel represents a 2-D histograms of MAGARA vs AERONET 500 nm fine-mode AOD. The lower-right panel represents a 2-D histograms of MAGARA vs AERONET 500 nm coarse-mode AOD. Statistics for each panel are presented in the title. For the comparisons of AOD, a standard +/- (0.03 + 0.15 * AOD) uncertainty envelope is provided for reference, with the percent meeting that threshold indicated**





### 4.3 MAGARA single-scattering albedo comparison with AERONET

Section 3 provides evidence that MAGARA should have (Figure 6) some sensitivity to single-scattering albedo over land. Figures 5, 7, and 9 provide some evidence that MAGARA probably has qualitative sensitivity to single-scattering albedo. Additionally, Figure 8 indicates that MAGARA may also have sensitivity to over-land gradients of smoke plume SSA. Here, 785 we perform a statistical comparison of MAGARA retrieved SSA against AERONET retrievals of single-scattering albedo performed during almucantar inversions. As in the prior two subsections, we perform 9x9 pixel averages of spectral AOD and spectral absorbing AOD before converting to spectral single-scattering albedo. AERONET spectral fine-mode, coarse-mode, and absorbing AOD are temporally averaged over +/- 10-15 minutes. Still, because the almucantar inversions are done


only every hour (at best), we choose the closest (temporal) AERONET inversion. Because AERONET data are unavailable for MAGARA's longest wavelengths, we only compare results for the following wavelengths: 470 nm, 550 nm, 636 nm, and 864 nm. AERONET spectral AOD (fine, coarse, and absorbing) data are interpolated to these wavelengths via fitting to a 2$^{nd}$ order polynomial in log-log space prior to conversion to single-scattering albedo.

     It is important to note for this analysis that retrievals of AERONET single-scattering albedo are not fully spectrally independent, as restrictions on the smoothness of the imaginary part of refractive index can cause artificial biases in the

retrieved spectral single-scattering albedo (*Sinyuk et al.*, 2022). Because MAGARA employs discrete aerosol optical models, the same can be said for MAGARA results. Additionally, MAGARA only retrieves fine-and-coarse-mode aerosol particle properties at a daily cadence. However, because fine-mode fraction is allowed to vary at the cadence of reported AOD, MAGARA single-scattering albedo can vary over the course of the day. We present color-coded scatterplots of MAGARA and AERONET spectral single-scattering albedo as a four-panel plot in Figure 13. For each panel except for the lower right,

statistics for all data are presented in the title. We also provide statistics for each spectral band separately in the color-coded legend. At lower AOD (upper left panel; $0.3 \leq$ MAGARA spectral AOD < 0.5), the 126 MAGARA retrievals show good agreement with AERONET: RMSE = 0.022, MAE = 0.018, and r = 0.752. Because data are only plotted here if MAGARA-retrieved **spectral** AOD falls within the range of 0.3-0.5, agreement improves with wavelength, as these data are mostly fine-mode dominated smoke plumes. At higher AOD (upper right panel; MAGARA spectral AOD > 0.5), correlation

continues to improve to 0.84, but the error statistics for the 116 MAGARA data points worsen substantially: RMSE = 0.030 and MAE = 0.028. If we plot MAGARA-AERONET spectral SSA as a function of MAGARA-retrieved spectral AOD (lower-right panel), we find the source of the discrepancy between correlation and error. Compared to AERONET, MAGARA spectral single-scattering albedo becomes increasingly negatively biased with increasing retrieved AOD. Although we are unsure of the cause here, a simple bias correction is sufficient to significantly mitigate the issue.

Figure 14 shows the same MAGARA/AERONET comparison, this time with the following MAGARA bias correction: SSA = SSA + $0.055 - 0.075e^{-\tau}$, where $\tau$ represents spectral aerosol optical depth. Additionally, MAGARA SSA is capped at 0.995, as this is a more realistic upper bound for spectral SSA. All spectral bands show large improvements in MAGARA SSA error statistics at elevated AOD (AOD > 0.5): RMSE drops 50% from 0.030 to 0.015, MAE drops nearly 65% from 0.028 to 0.01, and the correlation coefficient increases from 0.84 to 0.87.

A comparison of AERONET and MAGARA AAE was also performed, but is not included here. Although the agreement with AERONET was poor, there are only 14 MAGARA/AERONET coincidences with MAGARA 863 nm AOD > 0.5. Additionally, the angular smoothness constraints placed upon the AERONET retrieval could substantially impact AERONET's results (*Sinyuk et al.*, 2022; *Wagner and Silva*, 2008). AERONET Version 4 should address these issues, and we look forward to a comparison at that point.




**Figure 13) Over-land MAGARA vs AERONET spectral single-scattering albedo comparison, conditioned on MAGARA retrieved spectral AOD. Upper-left panel shows MAGARA (y-axis) vs AERONET (x-axis) spectral single-scattering albedo, for MAGARA retrieved AODs between 0.3 and 0.5. Upper-right panel shows MAGARA (y-axis) vs AERONET (x-axis) spectral single-scattering albedo, for MAGARA retrieved AODs greater than 0.5. Lower-left panel shows MAGARA (y-axis) vs AERONET (x-axis) spectral**
**single-scattering albedo, for MAGARA retrieved AODs greater than 0.3. Lower-right panel shows MAGARA single-scattering albedo errors (y-axis) vs MAGARA retrieved spectral AOD (x-axis), for MAGARA AOD > 0.3. Statistics for all points within a plot are presented in the title. Spectral single-scattering albedo statistics are presented in the legend. The solid black line in the lower-right panel is represented by $0.075e^{-AOD} - 0.055$.**



**Figure 14) Same as Figure 13, but with bias correction of (0.075e<sup>-AOD</sup> – 0.055) subtracted from the data.**

**Figure 14) Same as Figure 13, but with bias correction of $(0.075e^{-AOD} - 0.055)$ subtracted from the data.**

## 5 Conclusions

In Section 1, we delve into the history of geostationary remote sensing, briefly describing some of the advances in geostationary imagers, especially as they relate to the advanced baseline imager. We also present MAGARA in the context of other work being done in this area. Section 2 gives a detailed overview of MAGARA, from data download and initial processing to aerosol and surface retrievals. By the end of section 2, MAGARA should manifest as a multi-faceted pixel-level (up to 1 km) retrieval algorithm that operates on relatively simple and realistic (for certain geographic regions),





assumptions for a given location in the geostationary field-of-view: (1) initial cloud screening is adequate to remove most clouds; (2) the surface BRF (i.e., reflectance) changes minimally from day-to-day for a given time of day from a geostationary perspective; (3) fine- and coarse-mode aerosol particle properties vary minimally over the course of a day, though their AOD and fine-to-coarse ratio might not; and (4) aerosol loading (i.e., AOD) and the fraction of fine-to-coarse modes may change rapidly over 10-minute to 1-hour time-scales. Because the last three assumptions hold up quite well in

most situations, as illustrated by the results, the success of MAGARA for these current case studies is mostly driven by our ability to screen clouds and cloudiness in general, a common problem in all aerosol remote sensing.

In Section 3, we present three case studies demonstrating the qualitative efficacy of MAGARA: the 2018 Camp Fire, the 2019 Williams Flats Fire, and the 2019 Kincade Fire. Retrievals for the 2019 Camp Fire, which devasted the town of Paradise, California, demonstrate MAGARA's qualitative sensitivity to total aerosol loading, single-scattering albedo and

fine-mode fraction. These results are impressive considering that MAGARA was run only for GOES-16, as GOES-17 data were unavailable.

For our second case study, we analyzed the 2019 Williams Flats Fire over the Pacific Northwest region of the United States and southwestern Canada. AERONET-reported AODs are too low to validate information about aerosol particle properties. Still, retrievals of single-scattering albedo from MAGARA suggest qualitative sensitivity to gradients in

single-scattering albedo, consistent with MISR RA retrievals over the same region. This case study also highlights one of the problems with MAGARA, an assumption of near-invariance in the surface reflectance from day to day.

For our third case study, we analyzed the 2019 Kincade fire over the same region as for the Camp Fire case study. Over the course of several days, MAGARA observed several smoke plumes, identifying them as being dominated by fine-mode, absorbing aerosol. Interestingly, two days within this case study show significant dust plumes which MAGARA

retrieves as coarse-mode aerosols, one of which is also captured by AERONET. This case study also highlighted a potential limitation of MAGARA. A large georegistration anomaly on October 30th for GOES-16 caused the algorithm to mask out much of that days' retrievals as poor quality. Because MAGARA tiles observations over both day and time-of-day for several days, the algorithm is especially sensitive to temporally varying radiometric/georegistration errors.

In Section 4, we provide comprehensive validation of MAGARA AOD, fine-mode fraction (FMF), and single-

scattering albedo (SSA) against AERONET using only +/- 10-15 minute temporal averages for AERONET while also providing some context with NOAA's bias-corrected aerosol product. After identifying an optimal averaging distance of $9^2$ pixels (retrievals), we compare MAGARA and the NOAA bias-corrected product against AERONET for the same 8,443 coincidences. Overall, MAGARA median absolute error (MAE; 0.016 vs. 0.021) and root-mean-squared error (RMSE; 0.040 vs. 0.049) are approximately 25% lower than the NOAA bias-corrected retrievals, with a correlation coefficient of about

0.12 larger (r; 0.785 vs. 0.666). Additionally, MAGARA quality appears significantly better at smaller spatial scales (smaller averaging window).



Comparing MAGARA retrievals of fine-mode fraction to AERONET spectral deconvolution data (n=384), we report the following over-land error statistics: MAE=0.031, RMSE=0.100, and r=0.902. This suggests that MAGARA is sensitive to fine-mode fraction at a temporal cadence of 10-15 minutes.

We also compare retrievals of MAGARA spectral single-scattering albedo with AERONET, even though MAGARA only retrieves fine-and-coarse-mode aerosol particle properties at a daily cadence. Because MAGARA does retrieve fine-mode fraction at a much higher cadence, retrieved single-scattering albedo technically varies at the cadence of reported AOD/FMF. For MAGARA-retrieved spectral AOD between 0.3 and 0.5 (relatively low AOD, n=126), MAGARA agrees well with AERONET: MAE=0.18, RMSE=0.022, and r=0.752. At higher AOD, MAGARA suffers from a large

negative bias in retrieved SSA, resulting in the following error statistics for our 116 MAGARA/AERONET coincidences: MAE=0.028, RMSE=0.030, and r=0.840. A simple single parameter (AOD) bias correction is then presented, resulting in the following error statistics at high AOD: MAE=0.010, RMSE=0.015, and r=0.870. Although we don't delve into the causes of this SSA bias, it is likely at least partially because MAGARA only retrieves fine-and-coarse-mode aerosol properties at a daily cadence, and that the much higher cadence of retrieved fine-mode fraction is unable to compensate for real diurnal

variability in SSA. From a retrieval standpoint, the fact that an AOD-based bias correction can substantially mitigate this issue is very interesting and convenient. That being said, we will need much more data to verify these findings.

     This manuscript demonstrates MAGARA's ability to retrieve AOD and aerosol properties such as fine-mode fraction and single-scattering albedo. Satellite observations from the Camp Fire case indicate that aerosol absorption was substantial at 870 nm, as the top-of-atmosphere signal declined significantly when the plume moved over the region.

MAGARA was capable of discriminating smoke from dust for the Kincade Fire case study. MAGARA's ability to discern aerosol loading and particle properties from geostationary data could profoundly impact our ability to accurately model aerosol within climate models. Additionally, the 10-minute cadence of MAGARA retrievals and our ability to accurately separate the fine and coarse modes could significantly improve air-quality modeling and forecasting in certain regions. The regions most likely to benefit from a MAGARA style approach are those places where surface reflectance is slow to vary

and cloud-cover is minimal: the western US, North-Central Africa, the Middle East, parts of Central Asia, and large portions of Australia.

**Data availability.**

All MAGARA and NOAA GOES-16/17 bias-corrected data used for this manuscript will be published to a NASA and/or NOAA repository prior to final publication.

**Author contributions.**

MAGARA was designed and built by JAL under the supervision of RAK. MF and TS assisted with bug-fixing and documenting the MAGARA code. MF also ported RT software to NASA supercomputers to develop the first iteration of the



RT LUT used here. JL developed the dust aerosol optical model used by MAGARA. HZ developed the algorithm for the NOAA GOES-16/17 bias-corrected AOD and HZ ran the algorithm for the case studies presented herein. The manuscript is based on Chapter 4 of JAL's doctoral dissertation, with substantial edits for this submitted manuscript. This manuscript was written by JAL with advisement by RAK and input from all authors.

**Competing interests.**

The authors declare that that have no conflict of interest.

**Acknowledgments**

We thank Brent Holben and Pawan Gupta from NASA Goddard and the AERONET team for producing and maintaining this
critical validation dataset. We thank the NOAA groups responsible for calibrated radiances and the aerosol product. We thank Alexei Lyapustin and the MAIAC team for the MODIS MAIAC products used in this manuscript and for many years of useful insight. We thank Sergey Korkin for help with RT questions, Shana Mattoo for the ABI geometry tables, and Andrew Sayer for helpful discussion. We thank George Young and Eugene Clothiaux from Penn State University for constructive feedback. We also thank Drs. Rozanov and the SCIATRAN team for their work on the SCIATRAN product. Resources supporting the
RT runs were provided by the NASA High-End Computing (HEC) Program through the NASA Center for Climate Simulation at Goddard. This research was supported in part by NASA's Climate and Radiation Research and Analysis Program under Hal Maring, NASA's Atmospheric Composition Program under Richard Eckman, the EOS Terra project under Kurt Thome, NASA's New Investigator Program under Allison Leidner, and NASA's HEC Program under Tsengdar Lee.

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
