# Peer review of "MAGARA: A Multi-Angle Geostationary Aerosol Retrieval"

_Atmospheric Measurement Techniques, 2023_

## Author Comment (AC1)

Authors' response to comments are highlighted in **red.**

This paper proposes a new algorithm to retrieve aerosol optical depth, fine mode fraction, surface properties as well as fine-and-coarse aerosol particle properties. The algorithm has been applied to three cases and evaluated with AErosol Robotic NETwork (AERONET). The results proves that the algorithm works well in retrieving in aerosol properties including AOD, single scattering albedo and fine mode fraction. The paper is well written. I would suggest to accept the paper after minor revisions. Specific comments are give below.

5

**The authors thank reviewer one for their feedback.**

10   1) The modules are looped through fixed numbers of times instead of optimal numbers of times. Does it serve for parallel programming? Could you explain how you determine the certain numbers the different modules are looped? Have you done any experiments to prove that there is really a need to have so many loops? I would like to know how long it takes to process a 50-by-50-pixel region?

**The loop numbers were determined experimentally with the Camp Fire case (only 1 imager available), and then were applied to all 3 case studies. We experimented with optimal looping structure (error < x), but results did not improve much and run time increased substantially. The algorithm is parallelized extremely efficiently, as each 50x50 (configurable) region is run independently on a separate core (if available). Because there is no pixel-to-pixel cross-talk for this algorithm, it is inherently well-suited for parallelization. A 50-by-50 pixel region (run over 2 weeks of data) will take no more than a few hours on a single core of a modern (post 2020) machine.**

15

2) In "Aerosol/Surface retrieval iteration" section of Fig. 3, "If iterInd==4" should be "If iterInd==5".

20 **It is correct as indicated, iterInd references the loop above (4 is the maximum value).**

3) In Fig. 5, the aerosol loading over water in some area (eg. upper left area on 24, 25, 27 October) should be large, while the retrieved AOD values are small. Could you give some explanation?

**We believe you mean Figure 9. This is a convolution of multiple issues.**

25     **1) The over-water surface appears to not be sufficiently well-characterized for these regions.**
            **a. Partially due to insufficient initial cloud-screening (in general a likely culprit for this algorithm).**
            **b. Probably partially due to variation (over multiple days) in 10-m wind, which could cause large changes in the TOA radiation field, even for a Rayleigh atmosphere.**
        **2) Cloud/Quality screening should probably have screened these higher AOD retrievals out.**
        **3) Logarithmic scale shows these artifacts much more prominently at lower AOD.**

30 **Using a Cox-Munk model in the future would probably alleviate these issues to some extent, as would improvement in quality masking. We have added these points to the discussion of Figure 9 in the text.**

4) Please add unit on x axis in Fig. 8.

**Units are already listed on the x-label (pixel).**

---

## Author Comment (AC2)

Authors' response to comments are highlighted in **red.**

Review of paper:

5 General comments:

This paper describes a pixel-level (up to 1 km) Multi-Angle Geostationary Aerosol Retrieval Algorithm that retrieves pixel-level aerosol loading and fine-mode fraction at up to the cadence of the measurements (10 minutes), fine-and-coarse mode aerosol particle properties at a daily cadence. Several case studies over the Desert Southwest, Pacific Northwest, and fire occurred regions are presented. The fine-mode AOD, coarse-mode AOD, and single-scattering albedo (SSA) of MAGARA

10 are compared to the AERONET and NOAA GOES-16 and GOES-17 products, which shows acceptable agreements. Aerosol type and loading of MAGARA at temporal resolution of 10 minutes are helpful for the new insights of aerosol-cloud interactions, improvements of air-quality modeling and forecasting, and additional constraints on direct aerosol radiative forcing. Therefore, the efforts on retrieval of detailed aerosol optical properties with high temporal resolution in this study are commendable and the work is meaningful. However, I have some comments on the current manuscript.

15 **The authors thank reviewer two for their comments. There have been many papers published on the retrieval of 550 nm AOD, so we hoped that offering a retrieval focusing on aerosol particle properties would be well-received.**

Major comments:

    1.   The abstract is too long and it needs to be further summarized.

**We have shortened the abstract a bit, but the paper details both a description of the algorithm and initial validation**
20 **of both AOD and aerosol particle properties (rather than just 550 nm AOD), so we need to leave enough detail in for readers to determine relevancy.**

    2.   Could you provide some comparisons of MAGARA products to geostationary Himawari-8/AHI products, at least for aerosol optical depth and angstrom exponent?

**We provide initial comparison against the NOAA ABI bias corrected AOD in the paper. The additional validation**
25 **suggested would be outside of the scope of this work. If there is sufficient interest from the community in this algorithm, it would be possible to work on a follow-on study involving either AHI or FCI.**

    3.   Did you run the MAGARA algorithm with some artificial data? How about the uncertainty of MAGARA retrievals? Could you provide some quantitative assessment?

**We did not run it with artificial data and the aerosol models were pulled from our previously published MISR work.**
30 **The uncertainty information we can provide comes from the comparison with AERONET, the best available and most-commonly used validation dataset for satellite aerosol retrievals. We agree this might not be too meaningful for a much broader dataset (certainly not pixel-level). Again, if there is sufficient community interest in this algorithm, further theoretical uncertainty studies and comparison with field-campaign data would be possible in the future.**

    4.   Could you provide the aerosol component retrievals in the MAGARA algorithm? I did not see any results about the
35       component retrievals except fine and coarse mode AOD, FMF, and SSA. In my opinion, the Table 1 describes the climatology of aerosol types, not aerosol component. So, if yes, I strongly recommend using "aerosol types" to replace "aerosol component" throughout the texts including the texts in the figures (Figure 2)

**The full dataset is available at https://doi.org/10.5281/zenodo.8164566. This dataset includes AOD, aerosol component fraction, cost function, modeled BRF, surface BRF, and a detailed description of each aerosol component. SSA, FMF, effective radius, etc. can all be calculated from the component fractions, extinctions, and particle properties. Aerosol type and aerosol component are used somewhat interchangeably in this manuscript and others that we and others have authored. We used aerosol component fraction, as this represents what the algorithm retrieves using the NNLS algorithm.**

Minor comments:

1. The texts in the maps are too small. Please improve it.

**We have increased the size of the text.**

2. I think two digits are enough for the statistics.

**Considering the variability in the measurements can be pretty small for some parameters such as SSA, we think 3 digits is appropriate here. We tried to make sure we didn't over interpret our results.**

3. Please provide the full name of the abbreviation when mentioned at the first time. For example, MAIAC in line 23, AOD in line 28, GRASP in line 101

**Corrected, thanks.**

---

## Author Response (AR3)

Authors' response to comments are highlighted in **red.**

5
**Editor**

**The authors thank the editor for their feedback.**

Based on the latest comments from RC2, I would like to ask the authors to further revise the paper:
10   1) Please shorten the abstract and leave out less important details.

**The abstract has been shortened from 513→335 words and is much more concise.**

2) Please include a quantitative assessment of MAGARA retrievals, which I believe it is critical for algorithm papers.

**In the paper, we have performed the following assessment of the MAGARA retrievals:**
15   **Section 3 & Figs. 5-9. Three wildfire cases analyzed in detail.**
**Section 4 & Figs. 10-14. Quantitative statistical comparison of AOD and particle properties with AERONET.**
**This represents a comprehensive validation against essentially all available data; there are no other data sources that can serve as validation for the MAGARA retrievals.**

20   3) In the conclusion, please describe the future work (satellite validations, algorithm improvement, etc.).

**We have added the following to the end of our conclusions:**

**Future work for MAGARA includes the ingestion of high-resolution column water vapor and ozone, which will allow us to better account for trace gas absorption.  Additionally, ABI 1.37-micron and thermal infrared channels will be utilized to improve cloud screening, allowing for the algorithm to be run in regions where cloudiness is more**
25   **pervasive.  Depending on the level of interest from the community, we may also consider extending MAGARA to other geostationary imagers in the future.**

**RC1**
30

This paper proposes a new algorithm to retrieve aerosol optical depth, fine mode fraction, surface properties as well as fine-and-coarse aerosol particle properties. The algorithm has been applied to three cases and evaluated with AErosol Robotic NETwork (AERONET). The results proves that the algorithm works well in retrieving in aerosol properties including AOD, single scattering albedo and fine mode fraction. The paper is well written. I would suggest to accept the paper after minor
35   revisions. Specific comments are give below.

**The authors thank reviewer one for their feedback.**

1)   The modules are looped through fixed numbers of times instead of optimal numbers of times. Does it serve for parallel programming? Could you explain how you determine the certain numbers the different modules are looped? Have you

done any experiments to prove that there is really a need to have so many loops? I would like to know how long it takes to process a 50-by-50-pixel region?

**The loop numbers were determined experimentally with the Camp Fire case (only 1 imager available), and then were applied to all 3 case studies. We experimented with optimal looping structure (error < x), but results did not improve much and run time increased substantially. The algorithm is parallelized extremely efficiently, as each 50x50 (configurable) region is run independently on a separate core (if available). Because there is no pixel-to-pixel cross-talk for this algorithm, it is inherently well-suited for parallelization. A 50-by-50 pixel region (run over 2 weeks of data) will take no more than a few hours on a single core of a modern (post 2020) machine.**

2) In "Aerosol/Surface retrieval iteration" section of Fig. 3, "If iterInd==4" should be "If iterInd==5".

**It is correct as indicated, iterInd references the loop above (4 is the maximum value).**

3) In Fig. 5, the aerosol loading over water in some area (eg. upper left area on 24, 25, 27 October) should be large, while the retrieved AOD values are small. Could you give some explanation?

**We believe you mean Figure 9. This is a convolution of multiple issues.**

    **1) The over-water surface appears to not be sufficiently well-characterized for these regions.**
        **a. Partially due to insufficient initial cloud-screening (in general a likely culprit for this algorithm).**
        **b. Probably partially due to variation (over multiple days) in 10-m wind, which could cause large changes in the TOA radiation field, even for a Rayleigh atmosphere.**
    **2) Cloud/Quality screening should probably have screened these higher AOD retrievals out.**
    **3) Logarithmic scale shows these artifacts much more prominently at lower AOD.**

**Using a Cox-Munk model in the future would probably alleviate these issues to some extent, as would improvement in quality masking. We have added these points to the discussion of Figure 9 in the text.**

4) Please add unit on x axis in Fig. 8.

**Units are already listed on the x-label (pixel).**

**RC2**

Review of paper:

General comments:

This paper describes a pixel-level (up to 1 km) Multi-Angle Geostationary Aerosol Retrieval Algorithm that retrieves pixel-level aerosol loading and fine-mode fraction at up to the cadence of the measurements (10 minutes), fine-and-coarse mode aerosol particle properties at a daily cadence. Several case studies over the Desert Southwest, Pacific Northwest, and fire occurred regions are presented. The fine-mode AOD, coarse-mode AOD, and single-scattering albedo (SSA) of MAGARA are compared to the AERONET and NOAA GOES-16 and GOES-17 products, which shows acceptable agreements. Aerosol type and loading of MAGARA at temporal resolution of 10 minutes are helpful for the new insights of aerosol-cloud interactions, improvements of air-quality modeling and forecasting, and additional constraints on direct aerosol radiative

forcing. Therefore, the efforts on retrieval of detailed aerosol optical properties with high temporal resolution in this study are commendable and the work is meaningful. However, I have some comments on the current manuscript.

**The authors thank reviewer two for their comments. There have been many papers published on the retrieval of 550 nm AOD, so we hoped that offering a retrieval focusing on aerosol particle properties would be well-received.**

Major comments:

1. The abstract is too long and it needs to be further summarized.

**We have shortened the abstract a bit, but the paper details both a description of the algorithm and initial validation of both AOD and aerosol particle properties (rather than just 550 nm AOD), so we need to leave enough detail in for readers to determine relevancy.**

2. Could you provide some comparisons of MAGARA products to geostationary Himawari-8/AHI products, at least for aerosol optical depth and angstrom exponent?

**We provide initial comparison against the NOAA ABI bias corrected AOD in the paper. The additional validation suggested would be outside of the scope of this work. If there is sufficient interest from the community in this algorithm, it would be possible to work on a follow-on study involving either AHI or FCI.**

3. Did you run the MAGARA algorithm with some artificial data? How about the uncertainty of MAGARA retrievals? Could you provide some quantitative assessment?

**We did not run it with artificial data and the aerosol models were pulled from our previously published MISR work. The uncertainty information we can provide comes from the comparison with AERONET, the best available and most-commonly used validation dataset for satellite aerosol retrievals. We agree this might not be too meaningful for a much broader dataset (certainly not pixel-level). Again, if there is sufficient community interest in this algorithm, further theoretical uncertainty studies and comparison with field-campaign data would be possible in the future.**

4. Could you provide the aerosol component retrievals in the MAGARA algorithm? I did not see any results about the component retrievals except fine and coarse mode AOD, FMF, and SSA. In my opinion, the Table 1 describes the climatology of aerosol types, not aerosol component. So, if yes, I strongly recommend using "aerosol types" to replace "aerosol component" throughout the texts including the texts in the figures (Figure 2)

**The full dataset is available at https://doi.org/10.5281/zenodo.8164566. This dataset includes AOD, aerosol component fraction, cost function, modeled BRF, surface BRF, and a detailed description of each aerosol component. SSA, FMF, effective radius, etc. can all be calculated from the component fractions, extinctions, and particle properties. Aerosol type and aerosol component are used somewhat interchangeably in this manuscript and others that we and others have authored. We used aerosol component fraction, as this represents what the algorithm retrieves using the NNLS algorithm.**

Minor comments:

1. The texts in the maps are too small. Please improve it.

**We have increased the size of the text.**

2. I think two digits are enough for the statistics.

**Considering the variability in the measurements can be pretty small for some parameters such as SSA, we think 3 digits is appropriate here. We tried to make sure we didn't over interpret our results.**

5      3. Please provide the full name of the abbreviation when mentioned at the first time. For example, MAIAC in line 23, AOD in line 28, GRASP in line 101

**Corrected, thanks.**